# Stability and Oracle Inequalities for Optimal Transport Maps between General Distributions

**Shubo Li**
Department of Statistics
The Pennsylvania State University
State College, PA 16803
skl6034@psu.edu

**Yizhe Ding**
Department of Statistics
The Pennsylvania State University
State College, PA 16803
yvd5212@psu.edu

**Lingzhou Xue**
Department of Statistics
The Pennsylvania State University
lzxue@psu.edu

**Runze Li**
Department of Statistics
The Pennsylvania State University
rzli@psu.edu

## Abstract

Optimal transport (OT) provides a powerful framework for comparing and transforming probability distributions, with wide applications in generative modeling, AI4Science and statistical inference. However, existing estimation theory typically requires stringent smoothness conditions on the underlying Brenier potentials and assumes bounded distribution supports, limiting practical applicability. In this paper, we introduce a unified theoretical framework for semi-dual OT map estimation that relaxes both of these restrictions. Building on sieved convex conjugate, our framework has two key contributions: (i) a new map stability bounds that holds without any second-order regularity assumptions on the true Brenier potentials, and (ii) an oracle inequality that cleanly decomposes the estimation error into statistical error, sieved bias, and approximation error. Specifically, our approximation error is measured in the $L^\infty$ norm rather than Sobolev norm in the existing results, aligning more naturally with classical approximation theory. Leveraging these tools, we provide statistical error of semi-dual estimators with mild and verifiable conditions on the true OT map. Moreover, we establish the first theoretical guarantee for deep neural network OT map estimator between general distributions, with Tanh network function class as an example.

## 1 Introduction

Optimal Transport (OT) provides a powerful framework for transforming one probability measure into another. Concretely, given two distributions $\mu$ and $\nu$ on $\mathbb{R}^d$, the OT problem seeks a map $T : \mathbb{R}^d \to \mathbb{R}^d$ that transports $\mu$ to $\nu$ while minimizing a cost function. Beyond its elegant mathematical foundation, OT has found broad applications in Ai4Science [39, 7, 29], computer vision [3, 23, 43, 12], and nonparametric statistical inference [10, 26, 24, 18], powering advances in scientific discovery and machine/deep learning. Despite its broad impact, rigorous theoretical guarantees for OT estimators under mild and realistic regularity conditions remain limited.

This paper aims to contribute to this direction by focusing on the dual-type OT map estimators, formally introduced in Section 2, which are based on the characterization of the optimal transport map as the gradient of a convex function (the so-called Brenier potential). The seminal work of [27] established minimax optimality for a wavelet-truncated dual-type OT estimator, but under restrictive conditions: $\mu$ is supported on $[0, 1]^d$ with a bounded density, $\nu$ is compactly supported, and the true

39th Conference on Neural Information Processing Systems (NeurIPS 2025).

OT map is both $\alpha$-Hölder smooth and strongly convex. These assumptions exclude many practical OT applications.

In parallel, Gunsilius [25] introduced an alternative analysis using the Poincaré inequality, a fundamental tool in PDEs and functional analysis. While still assuming compact and convex supports, this approach removed the need for Hölder smoothness or strong convexity of the Brenier potential. Building on this, [21] extended the theory to sub-exponential $\mu$, under $(\alpha, a)$-convexity and $(\beta, a)$-smoothness conditions on the Brenier potential. More recently, [20] relaxed the Poincaré inequality assumption to accommodate heavy-tailed $\mu$ and $\nu$, and eliminated the $(\alpha, a)$-convexity condition via a new sieved OT estimator, characterizing how tail thickness of $\mu$ and $\nu$ influences convergence.

We also note recent advances in plug-in type OT estimators, which directly compute the OT map between estimated distributions [10, 17, 34, 37, 5]. In parallel with these theoretical advances, significant efforts have focused on improving the computational efficiency of OT, including entropic regularization [15, 1, 22, 11, 19] and sum-of-squares formulations [35, 41]. Meanwhile, neural-network-based OT estimators have advanced rapidly, powering large-scale applications such as image generation and translation [33, 30, 4, 31, 12].

Despite these advances, a few core regularity assumptions on the true Brenier potential $\varphi_0$ and distributions still limit the applicability of dual-type estimators. To clarify these limitations, we decompose the overall estimation error into two components: statistical error and approximation error. The statistical error measures the discrepancy between the estimator and the best possible function in $\mathcal{F}$, reflecting the randomness from finite samples. The approximation error captures the gap between the true Brenier potential and its best approximation within $\mathcal{F}$, since we do not assume that the true Brenier potential $\varphi_0$ lies in $\mathcal{F}$. Existing analyses of each error component often rely on restrictive smoothness or convexity assumptions, which we discuss in more below.

First, bounding the statistical error in existing works requires a $(\beta, b)$-smoothness condition on $\varphi_0$, i.e., a polynomial bound on its Hessian. While this condition is verifiable under strong log-concavity of $\mu$ and $\nu$ [8, 14, 13, 28], its validity beyond this class remains unclear. Moreover, Proposition 3.12 of [20] shows that this smoothness requirement constrains $\nu$ to have tails no much heavier than those of $\mu$, excluding many practical scenarios, such as multivariate quantile [9, 10, 24], where we transport from $\mathbb{U}([0, 1]^d)$ to distributions with unbounded support, or transporting from a Gaussian (with exponential tails) to a Student's $t$-distribution (with polynomial tails).

Second, controlling the approximation error typically relies on $(\alpha, a)$-convexity of $\varphi_0$, i.e. a generalization of strong convexity, which similarly imposes constraints on the relative tail behavior of $\mu$ and $\nu$ [20]. For instance, OT map from an unbounded $\mu$ to a bounded $\nu$ fails this condition. This is especially problematic in generative modeling, where the OT map is learned from Gaussian noise to image data of bounded values [40, 30, 38, 32]. Moreover, existing approximation error bounds assume that both $\mu$ and $\nu$ have bounded support, [27, 21].

These limitations raise a fundamental question in broadening the scope of optimal transport:

*Can one bound the statistical and approximation errors of dual-type OT estimators for general distributions without such restrictive assumptions?*

In this work, we provide a positive answer to the question posed above. Our two core innovations are a novel map stability inequality and a refined oracle inequality. Building on these tools, we are the first to establish statistical and approximation error bounds for dual-type OT estimators under minimal regularity assumptions. Table 1 provides a side-by-side comparison of the regularity conditions required by our results versus those in the existing literature.

Table 1: Regularity assumptions on the true Brenier potential $\varphi_0$ and $\mu$ required to establish statistical and approximation error bounds ($\mathcal{E}_{stat}$ and $\mathcal{E}_{app}$) in prior work and in our results.

|  | Assumption on $\varphi_0$ | | Distributional Assumption | |
| --- | --- | --- | --- | --- |
|  | $(\beta, b)$-smooth (for $\mathcal{E}_{stat}$) | $(\alpha, a)$-convex (for $\mathcal{E}_{app}$) | Tail thickness of $\mu$ | Poincaré inequality on $\mu$ |
| [27] | $b = 0$ | $a = 0$ | Compact | Implicitly Used |
| [25] | Not Required | Not Required | Compact | Required |
| [21] | Required | Required | Sub-exponential | Required |
| [20] | Required | Required | General | Poincaré-type inequality |
| Ours | Not Required | Not Required | General | Not Required |

Specifically, we summarize our main contributions as follows:

**1. A new map stability inequality without $(\beta, b)$-smoothness.** Stability inequalities are key to linking OT estimation error to the semi-dual objective. Existing results rely on $(\beta, b)$-smoothness of $\varphi_0$. In Proposition 3.3, we derive a new inequality that removes this requirement, assuming only a well-behaved OT map (Assumption 3.2), which holds unconditionally when $\nu$ is bounded and under mild conditions for unbounded distributions.

**2. A novel oracle inequality without $(\alpha, a)$-convexity.** Proposition 3.4 provides an oracle inequality that avoids the $(\alpha, a)$-convexity assumption on $\varphi_0$ and decomposes the estimation error into statistical error, sieved bias, and approximation error. Unlike prior works [21, 20], which involve harder-to-analyze gradient terms like $\inf_{\varphi \in \mathcal{F}} \|\nabla\varphi - \nabla\varphi_0\|_{L^2(\mu)}$, our bound relies on a more tractable supremum error over a compact region $B$: $\inf_{\varphi \in \mathcal{F}} \|\varphi - \varphi_0\|_{L^\infty(B)}$, making it more suitable for analysis using classical approximation theory.

**3. First non-asymptotic bounds under minimal smoothness.** With our new map stability inequality, Theorem 3.6 establishes the first non-asymptotic statistical error bound for dual-type OT estimators requiring only a first-order condition on $\varphi_0$, without second-order smoothness assumptions.

**4. First theoretical guarantee for neural OT estimators.** With our new tools, we derive the first convergence guarantee for neural OT estimators using smooth networks. For tanh neural networks (TNNs), Theorem 3.7 shows that if $\varphi_0 \in C^\alpha(\mathbb{R}^d)$ for $\alpha \geq 2$, then the neural OT estimator $\nabla\hat{\varphi}_n$ between unbounded distributions satisfies:

$$\mathbb{E}\|\nabla\hat{\varphi}_n - \nabla\varphi_0\|_{L^2(\mu)}^2 \lesssim_{\log n} n^{-\frac{\alpha}{d+2\alpha}}. \tag{1}$$

Notably, this rate matches that of [25], but under significantly weaker assumptions: we do not require compact support for $\mu$ and $\nu$, nor a Poincaré inequality on $\mu$.

**Notations and Terminologies**  Throughout this paper, we use the following notations. For $\alpha \in \mathbb{R}$, $\lfloor\alpha\rfloor = \{x \in \mathbb{Z} : x < \alpha\}$. For $v \in \mathbb{R}^d$, we write $\|v\|_2$ for its $\ell_2$ norm and $\|v\|_\infty$ for its $\ell_\infty$ norm. For a matrix $A \in \mathbb{R}^{d_1 \times d_2}$, let $\|A\|_\infty := \max_{i,j}|A_{i,j}|$. If $A$ is symmetric, $\|A\|_{op}$ denotes its operator norm, $\lambda_{\min}(A)$ denotes its smallest eigenvalue. For a function $f : \mathbb{R}^d \to \mathbb{R}^m$ and a measure $\mu$, we write $\mu f$, $\mathbb{E}_\mu f(X)$, or $\int f\, \mathrm{d}\mu$ for its integral, and $\|f\|_{L^2(\mu)} := (\int \|f\|_2^2\, \mathrm{d}\mu)^{1/2}$ for the $L^2$ norm. For $f : \mathbb{R}^d \to \mathbb{R}$ and a set $B \subset \mathbb{R}^d$, we denote $\|f\|_{L^\infty(B)} := \sup_{x \in B}|f(x)|$.

We write $a \lesssim b$ to mean $a \leq Cb$ for some constant $C > 0$ independent of $a$ and $b$, and $a \lesssim_{\log n} b$ to omit polylogarithmic factors in $n$. We write $a \asymp b$ if $a \lesssim b$ and $b \lesssim a$. We use $a \vee b$ and $a \wedge b$ to denote the maximum and minimum of $a$ and $b$, respectively. For sequences $a_n$ and $b_n$, we write $b_n = \mathcal{O}(a_n)$ if there exists a constant $c > 0$ such that $|b_n| \leq c\, a_n$, and write $b_n = \mathcal{O}_{\log}(a_n)$ if there exists constants $c_1, c_2 > 0$ such that $|b_n| \leq c_1 a_n \log(n)^{c_2}$. We use $\mathbb{I}$ to denote indicator function.

For $x \in \mathbb{R}^d$, write $\langle x \rangle := 1 + \|x\|_2$. When $x \in \mathbb{R}$, we set $\langle x \rangle := 1 + |x|$. We define $\log_+(x) := \max\{\log x, 1\}$. A function $\varphi \in C^2$ is called $(\boldsymbol{\beta}, \mathbf{b})$-**smooth** if $\|\nabla^2\varphi(x)\|_{op} \leq \beta\langle x \rangle^b$ for all $x$, and $(\boldsymbol{\alpha}, \mathbf{a})$-**convex** if $\lambda_{\min}(\nabla^2\varphi(x)) \geq \alpha\langle x \rangle^a$.

A probability measure $\mu$ is $(\boldsymbol{\lambda}, \mathbf{K})$-**sub-Weibull** if for $X \sim \mu$, $\mathbb{P}(\|X\|_2 \geq t) \lesssim \exp(-(t/K)^\lambda)$ for all $t > 0$. We refer to such a distribution as $\boldsymbol{\lambda}$-**sub-Weibull**, or simply **sub-Weibull**.

## 2 Background on Optimal Transport

Let $\mu$ and $\nu$ be two probability measures on $\mathbb{R}^d$. A measurable map $T : \mathbb{R}^d \to \mathbb{R}^d$ is called a push-forward from $\mu$ to $\nu$, if for any measurable set $A \subset \mathbb{R}^d$, $\mu(T^{-1}(A)) = \nu(A)$. We denote this by $\nu = T_{\#}\mu$. In Monge's problem, the optimal transport map is defined as the push-forward that minimizes the transport cost from $\mu$ to $\nu$:

$$\min_{T:\mathbb{R}^d \to \mathbb{R}^d} \int_{\mathbb{R}^d} \|x - T(x)\|_2^2\, \mu(\mathrm{d}x), \quad \text{s.t.} \quad T_{\#}\mu = \nu, \tag{2}$$

However, Monge's problem may not admit a solution, for example, when $\mu$ is discrete and $\nu$ is continuous. A natural relaxation is the Kantorovich formulation, which considers couplings $\pi \in \Pi(\mu, \nu)$ with marginals $\mu$ and $\nu$:

$$\min_\pi \iint \|x - y\|_2^2\, \pi(\mathrm{d}x, \mathrm{d}y), \quad \text{s.t.} \quad \pi \in \Pi(\mu, \nu). \tag{3}$$

Kantorovich formulation is always feasible [42]. As a linear program, its semi-dual form reads:

$$\min_{\varphi \in L^1(\mu)} \left\{ \int \varphi \, \mathrm{d}\mu + \int \varphi^* \, \mathrm{d}\nu \right\} = \min_{\varphi \in L^1(\mu)} \mu\varphi + \nu\varphi^*. \tag{4}$$

where $\varphi^*$ is the convex conjugate (a.k.a. Legendre-Fenchel conjugate) of $\varphi$:

$$\varphi^*(y) = \sup_{x \in \mathbb{R}^d} \{\langle x, y \rangle - \varphi(x)\}, \quad \text{for all } y \in \mathbb{R}^d. \tag{5}$$

The equivalence between Monge's problem (2), Kantorovich's formulation (3), and its semi-dual form (4) was established in the seminal work of Brenier [6]:

**Theorem 2.1** (Brenier's Theorem). *Let $\mu$ and $\nu$ be probability measures on $\mathbb{R}^d$, both with finite second moments, and assume that $\mu$ does not assign mass to small sets. Then there exists a convex function $\varphi_0$ that uniquely (up to constants) solves (4). Moreover, $\nabla\varphi_0$ solves Monge's problem (2), and $(\mathrm{Id} \times \nabla\varphi_0)_{\#}\mu$ solves the Kantorovich formulation (3).*

Brenier's Theorem reveals that the OT map is the gradient of a convex function $\varphi_0$, called the *Brenier potential*. Furthermore, since $\varphi_0$ is the minimizer of (4), we can estimate it using empirical measures $\mu_n$ and $\nu_m$ of $\mu$ and $\nu$, respectively. For a chosen function class $\mathcal{F}$, we define the Brenier potential estimator and the associated dual-type OT estimator as:

$$\hat{\varphi}_{n,m} = \arg\min_{\varphi \in \mathcal{F}} \mu_n\varphi + \nu_m\varphi^*, \quad \text{with OT map estimator} \quad \nabla\hat{\varphi}_{n,m}. \tag{6}$$

However, the supremum over the entire space $\mathbb{R}^d$ in the convex conjugate $\varphi^*$ makes the optimization procedure unstable and sensitive. It also introduces theoretical challenges when analyzing the convergence of dual-type OT estimator $\hat{\varphi}_{n,m}$. To address this issue, [20] proposed the concept of a *sieved convex conjugate*, which restricts the supremum to a bounded region. Specifically, for some sieve radius $\tilde{R} > 0$, define:

$$\varphi^*_{\tilde{R}}(y) := \sup_{x : \|x\|_2 \leq \tilde{R}} \langle x, y \rangle - \varphi(x), \quad \text{for all } y \in \mathbb{R}^d. \tag{7}$$

Note that $\varphi^*(\nabla\varphi(x)) = \langle x, \nabla\varphi(x) \rangle - \varphi(x)$. By restricting $x$ to $B(0, \tilde{R})$, we equivalently restrict the domain of $\varphi^*_{\tilde{R}}(y)$ to the compact set $\nabla\varphi(B(0, \tilde{R}))$. This compactness facilitates us to analyze the estimation error of sieved OT estimator.

Given a sieve radius $\tilde{R}$, the corresponding *sieved estimator* is defined as

$$\tilde{\varphi}_{n,m} := \arg\min_{\varphi \in \mathcal{F}} \mu_n\varphi + \nu_m\varphi^*_{\tilde{R}}, \quad \text{with sieved estimator} \quad \nabla\tilde{\varphi}_{n,m}. \tag{8}$$

Note that when $\tilde{R} = \infty$, we recover the original dual-type estimator in Equation (6). In the following, we may abbreviate it to $\tilde{\varphi}_n$ when $n = m$.

# 3 Main Results

In Sections 3.1 and 3.2, we develop two core analytical tools: a new stability inequality for the sieved estimator (Proposition 3.3), and a refined oracle inequality that cleanly decomposes the estimation error (Proposition 3.4). Building on these results, Section 3.3 establishes non-asymptotic statistical error rates for general function classes (Theorem 3.6). Finally, in Section 3.4, we show how the overall estimation error of neural OT estimators between general distributions can be controlled, using tanh-activated neural networks a concrete example (Theorem 3.7).

## 3.1 A New Map Stability Inequality

From a statistical learning perspective, map stability inequalities are essential for linking the semi-dual objective to estimation error. For the original dual-type estimator, denote its objective is $\mathcal{S}_{\mu,\nu}(\varphi) := \mu\varphi + \nu\varphi^*$. Existing results [27, 21, 20] typically show that for all $\varphi \in \mathcal{F}$,

$$\|\nabla\varphi - \nabla\varphi_0\|^2_{L^2(\mu)} \lesssim \mathcal{S}_{\mu,\nu}(\varphi) - \mathcal{S}_{\mu,\nu}(\varphi_0). \tag{9}$$

where $\nabla\varphi_0$ is the true OT map to be estimated.

However, these results apply only to the original estimator (6), not the sieved version in (8), which often performs better in practice. Moreover, while neither side of the inequality involves second-order derivatives, prior analyses still require $(\beta, b)$-smoothness of $\varphi_0$, limiting their generality.

To accommodate sieved estimator and relax their assumptions, we consider the following conditions:

**Assumption 3.1** (Envelope for Function Class). *Let $\mathcal{F} \subset C^2(\mathbb{R}^d)$ be a function class. We assume there exists a non-decreasing and pointwise finite function $U_2 : [0, \infty) \to [1, \infty)$ such that:*

$$\sup_{\varphi\in\mathcal{F}} \sup_{\|x\|\leq R} \|\nabla^2\varphi(x)\|_{op} \leq U_2(R), \qquad \sup_{\varphi\in\mathcal{F}} \|\nabla\varphi(0)\|_2 \leq U_2(R).$$

Assumption 3.1 depends only on $\mathcal{F}$ and is independent of $\varphi_0$, $\mu$ and $\nu$. It is satisfied by many common function classes, for example,

1. Quadratic function class: When $\mu$ and $\nu$ are both Gaussian (or from the same elliptical family), the Brenier potential is quadratic and lies in the following quadratic family:

$$\mathcal{F}_{quad} = \{x \mapsto x^\top B x + \langle b, x \rangle : B \in \mathbb{S}_+^d, \|B\|_{op} \leq r_1, \|b\|_2 \leq r_2\}.$$

2. Smooth Neural Networks: Neural network function classes with all parameters bounded and activation functions that are $C^2$-smooth, such as sigmoid, tanh, or softmax.

3. Reproducing Kernel Hilbert Spaces (RKHS): Let $K : \mathcal{X} \times \mathcal{X} \to \mathbb{R}$ be a positive-definite kernel on its domain $\mathcal{X} \times \mathcal{X}$, and let $\mathcal{H}_K$ be the corresponding RKHS with norm $\|\cdot\|_{\mathcal{H}_K}$. We assume $\mathcal{F}$ is $C^4$ and take $\mathcal{F}$ to be the unit ball of $\mathcal{H}_K$, i.e., $\mathcal{F} = \{\varphi \in \mathcal{H}_K : \|\varphi\|_{\mathcal{H}_K} \leq 1\}$.

Assumption 3.1 and 3.2 together play a role similar to Assumption A.1 in [21], but are strictly weaker. As a result, the function classes discussed in Section 4 of [21], including parametric family, wavelet expansions, RKHS, and Barron Spaces, apply directly to our setting. We refer readers to Section 4 of [21] for a detailed discussion.

**Assumption 3.2** (Envelope for OT map). *There is a function $u \in L^4(\mu)$, such that for all $\varphi \in \mathcal{F}$,*

$$\|\nabla\varphi(x) - \nabla\varphi_0(x)\|_2 \leq u(x), \quad \text{for all } x.$$

*Define $U_1(R) := \sup_{\|x\|_2 \leq R} u(x)$. Additionally, let $\Phi \in L^2(\mu)$ be an envelope function for $\mathcal{F} \cup \{\varphi_0\}$, i.e.*

$$|\varphi(x)| \leq \Phi(x), \quad \text{for all } x \in \mathbb{R}^d \text{ and all } \varphi \in \mathcal{F} \cup \varphi_0.$$

A sufficient condition for Assumption 3.2 is that both Proposition A.2 and Assumption 3.1 hold. Specifically, Proposition A.2 ensures $\|\nabla\varphi_0(x)\|_2$ to be bounded by some function $L_1(x)$. Meanwhile, Assumption 3.1 provides an envelope for $\|\nabla\varphi(x)\|_2$, denoted as $L_2(x)$. Then Assumption 3.2 follows via the triangle inequality $\|\nabla\varphi(x) - \nabla\varphi_0(x)\|_2 \leq L_1(x) + L_2(x)$. While Assumption 3.1 solely relies on $\mathcal{F}$, typical distributions for Proposition A.2 to hold include: $\mu$ can be normal or Student-t, and $\nu$ can be any measure with certain tail decay. We refer to Appendix A for further details and an explicit example verifying Assumption 3.2.

We are now ready to present our new stability inequality for the sieved estimator:

**Proposition 3.3** (Map Stability Inequality). *Suppose Assumptions 3.1 and 3.2 hold. For any $\varepsilon > 0$, define $R_\varepsilon$ and sieve radius $\tilde{R}_\varepsilon$ such that*

$$\mu(\|X\|_2 > R_\varepsilon) \leq \varepsilon, \quad \tilde{R}_\varepsilon \geq \sup_{\|x\|_2 \leq R_\varepsilon} \left\{\|x\|_2 + \frac{u(x)}{U_2(\|x\|_2 + u(x))}\right\} \tag{10}$$

*For any $\varphi \in \mathcal{F}$, define the truncated excess risk:*

$$r_\varepsilon(\varphi) := \int_{B(0, R_\varepsilon)} \varphi(x) + \varphi_{\tilde{R}_\varepsilon}^*(\nabla\varphi_0(x)) \, \mu(dx) - \int_{B(0, R_\varepsilon)} \varphi_0(x) + \varphi_0^*(\nabla\varphi_0(x)) \, \mu(dx). \tag{11}$$

*Then the following inequality holds: for any $\varphi \in \mathcal{F}$,*

$$\|\nabla\varphi - \nabla\varphi_0\|_{L^2(\mu)}^2 \leq 2U_2\big(R_\varepsilon + U_1(R_\varepsilon)\big) \cdot r_\varepsilon(\varphi) + \|u\|_{L^4(\mu)}^2 \cdot \varepsilon^{\frac{1}{2}}, \tag{12}$$

We first clarify the roles of $R_\varepsilon$ and $\tilde{R}_\varepsilon$, which enable a more tractable approximation error in the next section. Specifically, the hyper-balls $B(0, R_\varepsilon)$ and $\nabla\varphi_0(B(0, R_\varepsilon))$ serve as bounded pseudo-supports for $\mu$ and $\nu$, respectively, over which the behavior of $\varphi$ and its convex conjugate can be effectively controlled. Meanwhile, the sieve radius $\tilde{R}_\varepsilon$ is introduced to handle any mismatch between $\nabla\varphi\big(B(0, R_\varepsilon)\big)$ and the pseudo-support of $\nu$, by better regulating the range of the sieved conjugate. In practice, if $\nabla\varphi$ closely approximates $\nabla\varphi_0$ in $L^\infty$ norm (so that $u \approx 0$), the mismatch is negligible and one can take $\tilde{R}_\varepsilon \approx R_\varepsilon$. In that case, it suffices to compute the convex conjugate over the pseudo-support of $\mu$ alone.

For simplicity, a sufficient choice for $\tilde{R}_\varepsilon$ is $\tilde{R}_\varepsilon = R_\varepsilon + U_1(R_\varepsilon)$. In particular, if $\mu$ is supported on $B(0, R)$, one may take $R_\varepsilon = R$.

Equation (12) decomposes the estimation error into two components. The first arises from the truncated excess risk over the high-probability region $B(0, R_\varepsilon)$, which can be well-controlled (see Proposition 3.4). Its coefficient $U_2(R_\varepsilon + U_1(R_\varepsilon))$ originates from a second-order Taylor expansion of $\varphi$, capturing the "uniform curvature" of $\mathcal{F}$ on the ball $B(0, R_\varepsilon + U_1(R_\varepsilon))$. The second component accounts for the residual risk outside this region. As a result, $\varepsilon$ must be carefully selected to balance the trade-off: a smaller $\varepsilon$ reduces the residual risk but increases the truncated excess risk.

We defer the proof of Proposition 3.3 to Appendix B.

**Comparison with existing map stability results.** The most closely related results are Proposition 10 of [27], Proposition 1 of [21], and Lemma 3.15 of [20], all of which require $(\beta, b)$-smoothness assumptions on the true Brenier potential $\varphi_0$ and the candidate class $\mathcal{F}$. In contrast, our Proposition 3.3 substantially relaxes these requirements by assuming only an upper bound on the true OT map $\nabla\varphi_0$ via function $u(x)$.

Another key distinction is the introduction of the sieved convex conjugate $\varphi^*_{\tilde{R}_\varepsilon}$, essential for providing a more accessible approximation error in the next section. This sieving remains necessary even when $\mu$ has bounded support, unless additional regularity conditions are imposed on $\mu$ and $\nu$.

Compared to map stability results for plug-in type estimators, our bound also holds under milder assumptions. For instance, Theorem 2.1 of [17] requires $\nabla\varphi_0$ to be Lipschitz, while Theorem 3 of [5] assumes that the Brenier potential is strongly convex.

## 3.2 A New Oracle Inequality

Now, we present our second contribution of a new oracle inequality that helps cleanly decompose the estimation error into statistical error, sieved bias, and approximation error.

**Proposition 3.4** (Oracle Inequality). *Under Assumptions 3.1 and 3.2, and let $R_\varepsilon, \tilde{R}_\varepsilon$ be as defined in Equation* (10)*. For the sieved OT estimator $\tilde{\varphi}_{n,m}$ from Equation* (8)*, its truncated excess risk $\tilde{r}_{n,m,\varepsilon} := r_\varepsilon(\tilde{\varphi}_{n,m})$, defined in Equation* (11)*, admits the decomposition:*

$$\tilde{r}_{n,m,\varepsilon} := r_\varepsilon(\tilde{\varphi}_{n,m}) \leq \mathcal{E}_{stat} + \mathcal{E}_{sieve} + \mathcal{E}_{app}, \tag{13}$$

*where*

$$\mathcal{E}_{stat} := \sup_{f \in \bar{\mathcal{F}}} \int_{B(0, R_\varepsilon)} f \, d(\mu_n - \mu) + \sup_{g \in \bar{\mathcal{G}}} \int_{B(0, R_\varepsilon)} g \, d(\nu_m - \nu), \tag{14a}$$

$$\mathcal{E}_{sieve} := 2\mathbb{E}_{X \sim \mu_n}[\Phi(X) \cdot \mathbb{I}(\|X\|_2 > R_\varepsilon)] + 2\mathbb{E}_{Y \sim \nu_m}[G(Y) \cdot \mathbb{I}(\|(\nabla\varphi_0)^{-1}(Y)\|_2 > R_\varepsilon)], \tag{14b}$$

$$\mathcal{E}_{app} := 2 \inf_{\varphi \in \mathcal{F}} \|\varphi - \varphi_0\|_{L^\infty(B(0, \tilde{R}_\varepsilon))}. \tag{14c}$$

*Here, $\bar{\mathcal{F}} := \{\varphi_1 - \varphi_2 : \varphi_1, \varphi_2 \in \mathcal{F}\}$, $\bar{\mathcal{G}} := \{\varphi^*_{1,\tilde{R}_\varepsilon} - \varphi^*_{2,\tilde{R}_\varepsilon} : \varphi_1, \varphi_2 \in \mathcal{F}\}$ and $G(y) := \tilde{R}_\varepsilon\|y\|_2 + \sup_{x \in B(0, \tilde{R}_\varepsilon)} \Phi(x)$.*

Thus, with a careful choice of $\varepsilon$ and $R_\varepsilon$ (which can be selected in a distribution-free manner, as discussed in Section 3.3), Propositions 3.3 and 3.4 together provide a complete characterization of how the sample sizes, function class $\mathcal{F}$, and the sieved convex conjugate collectively influence the estimation error of the sieved dual-type estimators.

Specifically, this result decomposes the truncated excess risk into three components. The first term, $\mathcal{E}_{stat}$, reflects the statistical error from replacing the underlying distributions $\mu$ and $\nu$ with their

estimates $\mu_n$ and $\nu_m$. Importantly, these estimates need not be empirical measures, allowing for smoothed or bootstrap-based alternatives. The second term, $\mathcal{E}_{sieve}$, accounts for the bias introduced by replacing the original convex conjugate (Equation (7)) with its sieved version. This bias vanishes when $R_\varepsilon = \infty$, where the sieved and original convex conjugates coincide.

The third term, $\mathcal{E}_{app}$, captures the approximation error of the function class $\mathcal{F}$ in representing the true potential $\varphi_0$ over the compact domain $B(0, \tilde{R}_\varepsilon)$. As usual, enlarging $\mathcal{F}$ reduces $\mathcal{E}_{app}$ at the cost of increasing $\mathcal{E}_{stat}$. Notably, we express the approximation error in the $L^\infty$ norm on a compact set, aligning naturally with classical approximation theory. In contrast, prior works [27, 21, 20] typically rely on gradient-based errors like $\inf_{\varphi \in \mathcal{F}} \|\nabla \varphi - \nabla \varphi_0\|_{L^2(\mu)}$, which remain challenging to analyze for rich function classes such as deep neural networks.

**Proof ideas of Propositions 3.4** The sieve technique allows us to treat $B(0, R_\varepsilon)$ and $\nabla \varphi_0 \big(B(0, \tilde{R}_\varepsilon)\big)$ as bounded pseudo-supports of $\mu$ and $\nu$, or equivalently, as bounded pseudo-domains for $\varphi_0$ and $\varphi_0^*$. This reduction localizes the approximation problem to compact sets, where classical approximation theory applies. Another key benefit of this sieving approach is that it eliminates the need for the $(\alpha, a)$-convexity assumption required to control the approximation error in prior works.

The sieve bias is then controlled via tail probability bounds, while the statistical error is handled using standard tools from empirical process theory. The full proof is deferred to Appendix C.

## 3.3 Non-Asymptotic Bounds of Statistical Error and Sieved Bias

In this section, we derive bounds for the statistical error $\mathcal{E}_{stat}$ and the sieved bias $\mathcal{E}_{sieve}$ in Proposition 3.4, under minimal assumptions on the source distribution $\mu$. Notably, our results do not require smoothness of the true OT map $\nabla \varphi_0$, improving upon existing analyses.

We begin with the following assumption on the covering entropy of the function class $\mathcal{F}$:

**Assumption 3.5** (Covering Entropy of $\mathcal{F}$). *For some $\eta \geq 0$, $\gamma \in [0, 2)$, $\gamma' \geq 1$ and $D_\mathcal{F} > 1$, the covering entropy of $\mathcal{F}$ satisfies: for any $h \geq 0$,*

$$\log \mathcal{N}(h, \mathcal{F}, L^\infty([-R, R]^d)) \leq D_\mathcal{F} \cdot h^{-\gamma} \cdot \log_+(1/h)^{\gamma'} \cdot R^\eta. \tag{15}$$

We focus on the Donsker regime ($\gamma \in [0, 2)$) for theoretical simplicity, which already captures a wide range of function classes, such as parametric families, wavelets, reproducing kernel Hilbert space (RKHS), and both shallow and deep neural networks. Here, $D_\mathcal{F}$ represents the "effective dimension" of $\mathcal{F}$, and $n/D_\mathcal{F}$ can be viewed as the effective sample size from the statistical learning theory perspective. Because most covering entropy results are stated for the $L^\infty([0, 1]^d)$ norm, we introduce an additional $R^\eta$ factor to translate them to our unbounded setting in $L^\infty([-R, R]^d)$. Specifically, the exponent $\eta$ measures how the envelope of $\mathcal{F}$ grows (e.g. $\eta = 2$ if the envelope scales like $x \mapsto x^2$).

**Theorem 3.6** (Statistical Error with Empirical Measures). *Under assumptions of Proposition 3.4, and Assumption 3.5, the sieved estimator $\tilde{\varphi}_{n,m}$ from Equation (8), computed with empirical measures $\mu_n$ and $\nu_m$, satisfies:*

$$\mathbb{E}\Big[\mathcal{E}_{stat}\Big] \lesssim \sqrt{\frac{D_\mathcal{F}}{n}}\Big(R_\varepsilon^{\frac{\eta}{2}} + \sup_{x \in B(0, R_\varepsilon)} \Phi(x)^{\frac{1}{2}}\Big) \log(n)^{\frac{\gamma'}{2}} + \sqrt{\frac{D_\mathcal{F}}{m}}\Big(\tilde{R}_\varepsilon^{\frac{\eta}{2}} + M^{\frac{1}{2}}\Big) \log(m)^{\frac{\gamma'}{2}}, \tag{16}$$

*and*

$$\mathbb{E}\Big[\mathcal{E}_{sieve}\Big] \leq 2(\|\Phi\|_{L^2(\mu)} + \|G\|_{L^2(\nu)}) \cdot \varepsilon^{\frac{1}{2}}, \tag{17}$$

*where $M := \tilde{R}_\varepsilon \sup_{x \in B(0, R_\varepsilon)} \|\nabla \varphi_0(x)\|_2 + \sup_{x \in B(0, \tilde{R}_\varepsilon)} \Phi(x)$, and the suppressed constant depends only on $\gamma, \gamma'$.*

We defer the proof to Appendix D, where we also establish corresponding high-probability guarantees.

In Theorem 3.6, we establish a standard convergence rate for estimation error, $\mathbb{E}[\mathcal{E}_{stat}] = \mathcal{O}_{\log}(\sqrt{D_\mathcal{F}/n} + \sqrt{D_\mathcal{F}/m})$, which is common in empirical process theory. Meanwhile, the multiplicative prefactors in $\mathcal{E}_{stat}$, $R_\varepsilon^{\frac{\eta}{2}} + \sup_{x \in B(0, R_\varepsilon)} \Phi(x)^{\frac{1}{2}}$ and $\tilde{R}_\varepsilon^{\frac{\eta}{2}} + M^{\frac{1}{2}}$, naturally capture the "diameter" of $\mathcal{F}$ on the pseudo-supports of $\mu$ and $\nu$.

Moreover, it tells that the sieve bias $\mathcal{E}$ is governed by the tail probability $\varepsilon$ defined in Equation (10).

**Practical choice of $R_\varepsilon$ and $\tilde{R}_\varepsilon$.** As shown in [20], $R_\varepsilon$ can be selected in a distribution-free manner. Here, we extend this idea to provide a practical, joint selection strategy for both $R_\varepsilon$ and $\tilde{R}_\varepsilon$:

$$R_\varepsilon = \max_{1 \le i \le n} \|X_i\|_2, \qquad \tilde{R}_\varepsilon = R_\varepsilon + C \max_{1 \le j \le m} \|Y_j\|_2, \tag{18}$$

where $C \ge 0$ is a tuning parameter designed to account for the potential mismatch between the estimated and true OT maps, specifically the discrepancy $\|\nabla\tilde{\varphi}_{n,m} - \nabla\varphi_0\|_{L^\infty(B(0,R_\varepsilon))}$.

The choice of $R_\varepsilon$ can also be justified by our stability and oracle bounds (Propositions 3.3 and 3.4). In particular, if $n = m$, $\mu$ is sub-Weibull, and both $\Phi(\cdot)$, $\|\nabla\varphi_0(\cdot)\|_2$ grow at most polynomially, then the estimation error of the sieved estimator reads:

$$\mathbb{E}\|\nabla\tilde{\varphi}_n - \nabla\varphi_0\|_{L^2(\mu)}^2 = \mathcal{O}_{\log}\left((n/D_\mathcal{F})^{-\frac{1}{2}} + \mathcal{E}_{app} + \varepsilon^{\frac{1}{2}}\right). \tag{19}$$

By exchangability of $X_i$'s, choosing $R_\varepsilon := \max_{1 \le i \le n} \|X_i\|_2$ yields $\varepsilon \asymp n^{-1}$, so the sieved bias $\varepsilon^{\frac{1}{2}}$ is dominated by the statistical error term $(n/D_\mathcal{F})^{-\frac{1}{2}}$.

While a complete theoretical justification for our choice of $\tilde{R}_\varepsilon$ remains an open question, numerical results in Section 4 provide strong empirical support: our sieved estimator consistently outperforms the original dual-type estimator of [27, 25, 21] across a range of settings.

### 3.4 Estimation Error of Neural OT Estimator

In this section, we derive the first non-asymptotic error bounds for dual-type OT map estimators parameterized by deep neural networks between general distributions.

ReLU networks, despite their popularity, lack the smoothness needed to recover the gradient of the estimated Brenier potential. Moreover, our analysis (Assumption 3.1) requires $\mathcal{F}$ to be $C^2$. Consequently, smooth activations are both practically and theoretically preferable.

While input convex neural networks (ICNNs) [2] exploit convexity of Brenier potential and have been applied at scale [33, 4, 31], their approximation theory remains underdeveloped. This gap limits rigorous control of approximation error in existing analyses of dual-type estimators [27, 25, 21, 20]. By contrast, our approach removes the convexity assumption on $\mathcal{F}$, allowing us to employ standard fully connected networks.

Accordingly, we adopt the tanh-activated fully-connected neural network (TNN) as our function class $\mathcal{F}$, due to its $C^\infty$ smoothness and non-asymptotic universal approximation guarantees [16]:

$$f(x) = \mathcal{L}_L \circ \sigma \circ \mathcal{L}_{L-1} \circ \cdots \circ \sigma \circ \mathcal{L}_1(x), \tag{20}$$

where $\sigma(x) = \frac{e^x - e^{-x}}{e^x + e^{-x}}$ is applied entrywise, and $\mathcal{L}_\ell(x) = A_\ell x + b_\ell$ for $\ell \in [L]$, where $A_l \in \mathbb{R}^{p_{l+1} \times p_l}, b_l \in \mathbb{R}^{p_l}$, with $p_1 = d, p_{L+1} = 1$.

We define the TNN class $\mathcal{F}$ with bounded parameters as:

$$\mathcal{F}(L, W, \kappa) := \{f \text{ of form } (20) : \|A_\ell\|_\infty \vee \|b_\ell\|_\infty \le \kappa, \quad p_l \le W \text{ for all } \ell \in [L]\}, \tag{21}$$

where $\kappa > 0$ is a truncation threshold used for technical convenience.

**Theorem 3.7** (OT Estimation via Tanh Neural Network). *Let $n = m$, and suppose $\mu$ is $(\lambda, K)$-sub-Weibull, and the true Brenier potential $\varphi_0 \in C^\alpha(\mathbb{R}^d)$ with $\alpha \ge 2$ is $(\beta, b)$-smooth. By setting $R_\varepsilon, \tilde{R}_\varepsilon$ according to Equation (10), there exists a deep TNN function class $\mathcal{F} := \mathcal{F}(L, W, \kappa)$ where $L, W, \kappa$ depend on $n, d, \alpha, \beta, b$ and are of order $3 \le L = \mathcal{O}(1), \quad W = \mathcal{O}(n^{\frac{d}{d+2\alpha}}), \quad \kappa = \mathcal{O}(n^{\frac{d(d+(\lfloor\alpha\rfloor+2)^2+4)+2}{2(d+2\alpha)}})$, such that the sieved estimator $\nabla\tilde{\varphi}_n$ with sieve radius*

$$\tilde{R}_n = C_{K,d,\alpha} \cdot (\log n)^{\frac{1}{\lambda}}, \quad \text{for some constant} \quad C_{K,d,\alpha} \ge 4K\frac{2\alpha}{d+2\alpha} + 2 \tag{22}$$

*satisfies*

$$\mathbb{E}\|\nabla\tilde{\varphi}_n - \nabla\varphi_0\|_{L^2(\mu)}^2 \lesssim_{\log n} n^{-\frac{\alpha}{d+2\alpha}}. \tag{23}$$

**Remark 3.8** (On mild regularity conditions). *The sub-Weibull and $(\beta, b)$-smoothness assumptions are used in our work primarily for analytical convenience, not because they are fundamentally*

*required for the methodology. Notably, both conditions are widely used in the OT literature, see [21, 20, 27, 25, 34] for $(\beta, b)$-smoothness and [21, 17, 20] for sub-Weibull condition.*

*The $(\beta, b)$-smoothness of $\varphi_0$ is needed solely to control approximation error through Lemma F.4, ensuring the error grows polynomially in $\tilde{R}_\varepsilon$. Meanwhile, the sub-Weibull $\mu$ guarantees that $R_\varepsilon, \tilde{R}_\varepsilon$ scale polynomially in $\log(1/\varepsilon)$. Together, these two results ensure that the relevant Sobolev norm is bounded by a polynomial in $\log(1/\varepsilon)$, and the consequent approximation error has the order of $\varepsilon$.*

*If $(\beta, b)$-smoothness were lifted, one could still obtain approximation bounds, but they may degrade to $O(\varepsilon^a)$ for some $a < 1$. This complicates the statistical error, approximation error, and sieved estimation bias-variance trade-off by introducing additional convergence exponent. Likewise, relaxing source distribution to be polynomially tails causes $R_\varepsilon, \tilde{R}_\varepsilon$ to scale polynomially in $1/\varepsilon$. While the proofs follow similar steps, the resulting convergence rate becomes much more complex (See Equation 3.10 in [20] for an illustration of the added technical complications).*

**Remark 3.9** (On the choice of activation). *While ReLU networks are widely used in practice, our framework currently focuses on the tanh activation due to theoretical considerations. Specifically, our estimator is defined via the gradient of a potential function. Although PyTorch can handle ReLU networks computationally, the ReLU activation is not globally differentiable in mathematics, making this definition is ill-posed. Moreover, even under a weak derivative framework, ReLU networks lack second-order weak derivatives, whereas the existing optimal transport theory fundamentally relies on second-order regularity for statistical analysis.*

The rate in Theorem 3.7 matches that of [25], but under substantially milder assumptions. Although slower than the minimax rate in [27], this is expected: we do not assume $(\alpha, a)$-convexity of the true Brenier potential $\varphi_0$, compact support of $\mu$ and $\nu$, or a Poincaré inequality on $\mu$. Moreover, the estimators in [27, 25, 21] are either computationally intensive or NP-hard. In contrast, our deep TNN estimator is both implementable and scalable in practice.

We focus on the TNN architecture due to the current lack of approximation theory for general smooth neural networks. The proof of Theorem 3.7 is provided in Appendix E. With minor modifications, the result also holds with high probability.

# 4 Numerical Examples

In this section, we present numerical simulations to evaluate the performance of our sieved-TNN estimator and compare it against the original dual-type estimators from [27, 25, 21]. We consider four synthetic OT problems that pose challenges for existing theoretical frameworks:

- *Non-$(\beta, b)$-smooth maps:* $\quad \mathcal{N}(0,1) \to t_6, \quad \text{Uniform}(0,1) \to \mathcal{N}(0,1).$
- *Non-$(\alpha, a)$-convex maps:* $\quad t_6 \to \mathcal{N}(0,1), \quad \mathcal{N}(0,1) \to \text{Uniform}(0,1).$

We evaluate performance in dimensions $d = 5, 10, 20$, using i.i.d. samples $X_i \sim \mu$ and $Y_j \sim \nu$ with sample sizes $n = m \in \{64, 128, 256, 512, 1024, 2048\}$. Experiments were conducted on a server with Intel Xeon Gold 6342 processors, requiring approximately 4,000 CPU core hours in total.

**Implementation setup** We implement our estimator in PyTorch [36], following algorithms from [20] with objective function in Equation (8). The candidate class $\mathcal{F}$ consists of TNNs with two hidden layers of width 10 (for $d = 5$), 20 (for $d = 10$), and 30 (for $d = 20$). Training proceeds in two phases: a warm-up phase of $\max\{\lfloor 10000/n \rfloor, 50\}$ epochs at learning rate $5 \times 10^{-3}$, followed by 300 epochs at $10^{-3}$. Mini-batch size is 64 for both $\mu$ and $\nu$, and the sieved convex conjugate subproblem is solved using 300 inner iterations (see Algorithm 2 in [20]). In each trial, 10% of the samples are held out for validation. We defer implementation details to Appendix G.

**Sieve radius** As dicussed in Equation (18), we choose the sieve radius $\tilde{R}$ as

$$\tilde{R} = \max_{1 \le i \le n} \|X_i\|_2 \; + \; C \cdot \max_{1 \le j \le n} \|Y_j\|_2, \quad C \in \{0, 1, 2, 3, \infty\}.$$

Here, $C = 0$ corresponds to the setting in [20], while $C = \infty$ recovers the original dual-type estimator studied in [27, 25, 21], enabling direct comparison.

**Evaluation** We assess the estimation error using unexplained variance proportion (UVP) [30]: $L^2\text{-UVP}(\nabla\hat\varphi_n) := \|\nabla\hat\varphi_n - \nabla\varphi_0\|^2_{L^2(\mu)}/\mathrm{Var}_\nu(\|Y\|_2)$. Lower values indicate better performance. For each experiment, we approximate the $L^2$-UVP using an independent set of $10^6$ samples.

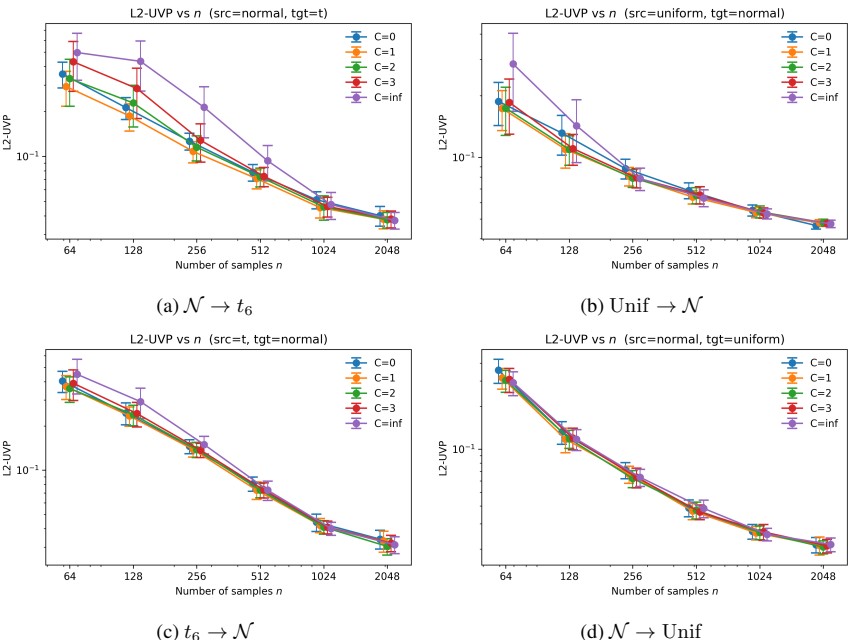

Figure 1: $L^2$-UVP when $d = 5$. Each curve shows the mean $L^2$-UVP over 50 random trials with one standard deviation.

Due to space constraints, we report only the $L^2$-UVP error versus sample size $n$ for $d = 5$ in Figure 1 here. The full set of numerical results is available in Appendix G.

Across all settings, our sieved TNN estimator consistently converges and outperforms the original dual-type estimator ($C = \infty$), particularly in the small-sample regime. As shown in Figure 4a, sieve-based estimators yield lower $L^2$-UVP errors for small $n$, while the classical estimator only catches up as $n$ grows large. This aligns with our theory: larger sieve radii may degrade convergence, and are only needed when the mismatching $\|\nabla\tilde\varphi_{n,m} - \nabla\varphi_0\|_{L^\infty}$, is large.

These results also offer practical guidance: a modest sieve constant (e.g., $C = 0$ or 1) suffices, and performance of our sieved estimator remains stable across a range of $C$.

## 5 Conclusion

In this paper, we develop new map stability and oracle inequalities for sieved dual-type OT estimators, without restrictive smoothness and convexity assumptions on the true Brenier potential $\varphi_0$ in the literature. Under mild regularity conditions, we establish the first non-asymptotic error bounds for neural OT estimators, using tanh networks as a concrete example. Numerical experiments further confirm the strong empirical performance of our sieved approach. Altogether, our unified framework advances the theoretical foundations of optimal transport and paves the way for future developments in both methodology and applications.

**Acknowledgment**: This research was partially supported the U.S. National Science Foundation under the grants DMS 2514400 and DMS 2210775, and by the U.S. National Institutes of Health under the grants R01GM163244 and 1R01GM152812.

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
