# OpenReview forum: "Stability and Oracle Inequalities for Optimal Transport Maps between General Distributions"
_NeurIPS.cc/2025/Conference — NeurIPS 2025 poster_

### Official Review · Reviewer_oQWg · 2025-06-16

**Clarity:** 3
**Significance:** 3
**Originality:** 3
**Rating:** 4
**Confidence:** 3

**Summary:**

The goal of this paper is to obtain OT estimators under relaxed assumptions: namely, relaxing assumptions on the measures (boundedness and/or convex support) and/or on the potentials themselves (e.g. global smoothness, strong convexity ...) (see Table 1).

To do so, the paper relies on an estimator introduced in [20] that is based on a modified semi-dual formulation of (quadratic) OT: instead a taking a global Legendre transform, the infimum is performed over a some ball; note that implicitly, this sieved estimator was also already considered in Vacher et. al. 2024 (Proposition 3.3). This sieved estimator enjoys stability properties under weaker assumptions than previous works (Proposition 3.3). Its total error is controlled in Proposition 3.4 and is made more explicit in the case of a low metric entropy (Theorem 3.6). As an application, they show that the sieved estimator used with an appropriate neural network provides an error which scales well with the smoothness of the true OT potential yet without the need to assume compactness, smoothness and strong-convexity for instance (Theorem 3.7).

**Questions:**

- Could Theorem 3.7 be obtained with [20]?
- Is there a simpler class than the neural network described in Theorem 3.7 that could "do the job"?

**Ethical Concerns:**

["NO or VERY MINOR ethics concerns only"]

**Final Justification:**

I maintain my score: I think the framework is nice and that Theorem 3.7 is novel.

**Limitations:**

- in the appendix, we can see that the network is actually quite large (exponential in the dimension). That would have been nice to say it explicitly somewhere.
- in the same vein, computing the sieve estimator remains a hard problem. That would have been nice to say it explicitly somewhere.
However, I fully understand that the contribution is not on the computational side hence it is a suggestion rather than a true limitation.

**Quality:**

3

**Strengths And Weaknesses:**

In my opinion, the main strength of the paper is Theorem 3.7, which I even found quite surprising to be honest.
The weaknesses are:
- it is not super clear that the sieved estimator is well-suited for practitioners: since the Legendre transform is restricted on a ball, the resulting sub-problem may be harder. However, I acknowledge that this is not the main point of the authors at all.
- on a technical level, it is not so clear to me how this work is not a marginal improvement of [20].

---

> ### Author Rebuttal · Authors · 2025-07-31
>
> We sincerely appreciate your thoughtful evaluation and constructive suggestions. Below, we respond to the identified weaknesses (W1-W2), limitations (L1-L2), and questions (Q1-Q2), and offer clarifications based on your valuable feedback.
>
> ## W1 & L2
>
> We first clarify the practical implementation of the sieved estimator, which we believe addresses your concerns regarding computational feasibility.
>
> The sieve estimator involves solving the inner optimization problem: for some $R\in(0,\infty]$ and function class $\mathcal{T}$, $$\varphi\_R^\ast(y)=\sup_{x\in\mathbb{R}^d:\|x\|_2\leq R}\langle x,y \rangle-\varphi(x).$$
> When $\varphi$ is convex (as is the case when using input convex neural networks, a common and practical choice in neural OT estimation), this is a convex optimization problem and can be efficiently solved using gradient descent. In fact, when $R = \infty$, corresponding to the classical Legendre transform, gradient-based methods have been effectively applied in the deep learning literature for years (e.g., [Hua+]).
>
> The sieved version simply introduces an additional projection step onto the $\ell_2$-ball of radius $R$ at each iteration of gradient descent. In another word, the computation of $\varphi_R^\ast$ can be achieved with projected gradient descent. As presented in our response to Q2 of mmaX, supplementary experiments show that the sieved estimator incurs only a modest computational overhead of about 15\% due to this projection. We emphasize that this moderate cost is justified by the consistent improvements in estimation accuracy and stability observed across all scenarios.
>
> Furthermore, we note that related works in neural OT estimation reformulate it as a min-max problem, e.g., let $\text{id}$ be the identity mapping, $$\hat\varphi_{n,m}=\min_{\varphi\in\mathcal{F}}\sup_{T\in\mathcal{T}}\mu_n \varphi+\nu_m(\langle \text{id}, T\rangle-\varphi\circ T), $$
> as seen in [Mak+, Tar+]. If the function class $\mathcal{T}$ is sufficiently rich and its outputs are uniformly bounded in $\ell_2$-norm by $R$, achievable via simple tricks like output truncation, then this formulation can be interpreted as another practical implementation of the sieve conjugate.
>
> In summary, while the sieved estimator offers theoretical advantages, it also remains computationally practical, incurring negligible additional burden for practitioners. Now, we move on to explaining your questions.
>
> ## W2 and Q1
>
> Thank you for this insightful question regarding the distinction between our results, especially Theorem 3.7, and related results in [DLX]. While our results may appear similar to [DLX], the two are fundamentally different in several key aspects:
>
> 1. **Characterization of Approximation Error.** A crucial conceptual difference lies in how the approximation error is quantified. In [DLX], it is expressed in terms of the excess risk functional $\inf_{\varphi} (\mu\varphi + \nu\varphi^\ast) - (\mu\varphi_0 + \nu\varphi_0^\ast)$, which does not directly connect with classical approximation theory tools. In contrast, we express approximation error in terms of the $L^\infty$ norm $\inf_{\varphi}\\|\varphi - \varphi_0\\|\_{L^\infty}$, making it easier to analyze using known approximation rates of TNNs.
>
> 2. **Flexibility in Sieve Radius Selection.** Our result accommodates a wide range of sieve radii $\tilde{R}\_\varepsilon$, allowing the choice of radius to be adapted based on the data and function class. In contrast, [DLX] fixes the sieve radius to the maximum empirical norm of the source sample, i.e., $\max_{1\leq i\leq n}\\|X_i\\|\_2$, and their proof techniques rely on this specific construction. It remains unclear whether this choice yields the same performance in our context, with TNNs as an example as the focus in our Theorem 3.7.
>
> 3. **Assumptions on Sample Sizes.** While we assume $n = m$ (equal sample sizes from $\mu$ and $\nu$) in Theorem 3.7 for theoretical analysis simplicity, this condition can be relaxed in our analysis (see Theorem 3.6). In contrast, Theorem 3.17 in [DLX] assumes $m \lesssim n$ due to their choice of sieve radius. This assumption appears restrictive and somewhat counterintuitive, as it limits the use of larger sample sizes from the target measure $\nu$, which should theoretically improve estimation performance. Upon closer inspection, we found this condition to be tightly coupled with the selection of sieve radius in [DLX], making it challenging to remove.
>
>
> While [DLX] provides valuable insights into the sieved estimator and presents appealing results on statistical error, their work does not address the analysis of approximation error. Without our novel techniques for selecting the sieve radius and for characterizing approximation error, it would not be possible to derive results such as Theorem 3.7. Our contributions fill this important gap, provide a unified framework to  capture both statistical error and approximation error in the estimation of OT maps.
>
> ## Q2
>
> Thank you for this thoughtful question regarding the function class used in Theorem 3.7. We focus on TNNs with smooth activation in Theorem 3.7 because this is an important setting in the literature where the approximation error is explicitly characterized in terms of the network’s complexity under smooth activation constraints. If one replaces TNNs with another nonparametric function class that is sufficiently expressive and satisfies the regularity assumptions in Assumptions 3.1 and 3.2, then analogous results could likely be established. Examples may include classical function spaces such as RKHS or wavelets, as studied in [DNP].
>
>
>
> ## L1
>
> Thanks for point this out. In our revised manuscript, we have explicitly listed the network architectures in Theorem 3.7, which is originally stated in Appendix E.
>
>
> ## Reference
>
> [DNP] Vincent Divol, Jonathan Niles-Weed, and Aram-Alexandre Pooladian. "Optimal transport map estimation in general function spaces." The Annals of Statistics 53.3 (2025): 963-988.
>
> [DLX] Ding, Li, and Xue. Statistical convergence rates of optimal transport map estimation between general distributions. ArXiv 2412.08064
>
> [Hua+] Huang et al. Convex potential flows: Universal probability distributions with optimal transport and convex optimization. ICLR 2021
>
> [Mak+] Makkuva et al. Optimal transport mapping via input convex neural networks. ICML 2020
>
> [Tar+] Tarasov et al. A statistical learning perspective on semi-dual adversarial neural optimal transport solvers. ArXiv 2502.01310

---

> > ### Comment · Reviewer_oQWg · 2025-08-03
> >
> > Thank you for your reply.
> > W1: ok thank you for providing the experimental computational overhead.
> > Q1: 1. Ok, this is indeed less redeable than your result.
> > 2. I mean, max of the norm seems like a reasonable choice of radius (or at least a multiple of it) but I understand that is weird that the analysis heavily depends on that particular choice.
> > 3. Ok I agree, it is also weird.
> > Q2: noted
> >
> > Overall I feel like the main contribution is the new stability result which I believe allows to close the gap in [20] but that’s fine by me for publication.

---

> > > ### Author Response · Authors · 2025-08-05
> > >
> > > Thank you for your insightful feedback.
> > >
> > > **Following up on Q1**
> > >
> > > We appreciate the reviewer’s thoughtful observations in Q1 and are glad to follow up on the three points.
> > >
> > > On **Q1(1)**, we’re grateful for the comment that our results are more readable. This clarity arises from our novel sieved estimator technique. Specifically, the sieved convex conjugate restricts the domain of $\varphi\_{\tilde{R}}^\*(y)$ to the compact set $\nabla\varphi(B(0,\tilde{R}))$, enabling us to analyze the estimation error of the sieved OT estimator. In contrast, [DLX] focuses solely on statistical error and does not provide an approximation error analysis.
> > >
> > > Regarding **Q1(2)**, we agree that setting the sieve radius to
> > > $\max_{1\leq i\leq n}\\|X_i\\|\_2$ is somewhat unnatural. A key strength of our framework is that it allows a flexible choice of $\tilde R_\varepsilon$, which enables principled trade-offs between approximation error, statistical error, and sieve bias.
> > >
> > > As for **Q1(3)**, we also agree that assuming $m \lesssim n$ is somewhat unnatural. Our theoretical results avoid such assumptions, thereby improving the transparency and interpretability of our theoretical guarantees.
> > >
> > > In summary, whereas [DLX] presents an analysis that isolates statistical error and leaves approximation behavior implicit, our framework offers a unified treatment of statistical and approximation error. This is enabled by new techniques for characterizing approximation error and for selecting the sieve radius in a principled way.
> > >
> > >
> > > **On the main contribution**
> > >
> > > We also wish to clarify that our main contribution lies not only in the new stability inequality (Proposition 3.3), but also in the oracle inequality (Proposition 3.4), statistical error bound (Theorem 3.6), and overall convergence for TNN (Theorem 3.7).
> > >
> > >
> > > Specifically, Proposition 3.3 shows that the $L^2$ estimation error of the OT map is controlled by a truncated excess risk and a residual tail risk, both formulated in terms of the Brenier potential. Proposition 3.4 then decomposes the truncated excess risk into three components: statistical error, approximation error, and sieved bias. Both Proposition 3.3 and 3.4 are designed for the sieved conjugate, which allows our analysis to focus on pseudo-compact supports and thus makes the study of the approximation error tractable.
> > >
> > > Building on Proposition 3.3, Theorem 3.6 establishes the statistical error bound under minimum smoothness assumptions on $\varphi_0$. By leveraging Propositions 3.3 and 3.4 together with the bound from Theorem 3.6, Theorem 3.7 ultimately derives the TNN approximation error.

---

### Official Review · Reviewer_qHh5 · 2025-06-18

**Clarity:** 2
**Significance:** 2
**Originality:** 3
**Rating:** 4
**Confidence:** 2

**Summary:**

The paper aims to study the statistical estimation of Optimal Transport (OT) maps.
Previous results required some balance between requiring compact support, or light tails, for the involved measures, and between requiring some possibly restrictive smoothness properties.
The present paper aims to lessen these assumptions by studying the so-called sieved estimator, which solves the dual problem when restricted to a bounded domain.
As a result of their analysis, the authors are able to derive new estimation rates for approximating the OT map using neural networks with Tanh activation.

**Questions:**

In line with my comments above, I have the following questions:

- How general are assumptions 3.1 and 3.2? Are there natural classes of measures, other than compactly supported, which satisfy these assumptions?
- Is there a *natural* application of Theorem 3.6 for maps with no a-priori smoothness guarantees?
- Theorem 3.6 will typically include dimensional factors. Can the authors explain the order of magnitude of the 'pre-factors' in Equation 18, under some proper normalization of the measures? The same thing would be useful for Equation 19.
- I found proposition 3.4 to also be interesting in the case R = infity. Can the authors comment on that case, when compared to previous results in the literature?
- Finally, it may be a bit naive on my side, but since Proposition 3.4 only depends on the Brenier potential, rather than the OT map itself, I hoped there could be a chance to consider less regular function classes, like ReLU networks. Can the authors comment on this point, and expand a bit on the discussion in Section 3.4?

**Ethical Concerns:**

["NO or VERY MINOR ethics concerns only"]

**Final Justification:**

I maintain my position that this is an interesting paper with potentially useful results.

**Limitations:**

No societal impact.

**Paper Formatting Concerns:**

The function M in Proposition A.2 has domain R, rather than R^d.

**Quality:**

3

**Strengths And Weaknesses:**

Strengths:
- Estimation of optimal transport is an interesting problem, and the results here should be interesting to the wider community.
- The results seem to be potentially quite general, which is important for applications.
- I particularly found the bound in Proposition 3.4 appealing. Expressing the error in terms of the Bernier potentials, rather than the OT map itself, seems like a useful thing.

Weaknesses:
- The paper is written in a very technical way, which sometimes makes it harder to synthesize the main ideas. For example, the way I see it, the use of the sieved estimator, rather than the standard dual estimator, should allow, in some sense, to reduce the problem into a compact support setting, where better results are known to hold. I think discussions of this sort and explanations of other crucial ideas can go a long way.
- The necessary assumptions (Assumptions 3.1 and 3.2) seem a bit opaque to me, and there is no actual discussion surrounding their necessity or roles. The authors do give an example in Appendix A, but this is a very specialized case. It would be interesting to see whether those assumptions can hold for heavier polynomial tails, and in general, it is important to explain the exact extent to which these assumptions will hold. As I see it, compactly supported measures are perhaps less interesting in the overall setting of this paper, so it's better to have a larger class, or at least easy-to-verify properties.
- As far as I understand, Theorem 3.7 is not algorithmic and requires searching over a very large function class. It's not a big disadvantage, since the paper mostly focuses on statistical perspectives, but I feel like some discussion about this point is in order. In particular, it would be good to explicitly state bounds on the overall complexity, depth and width.
- Moreover, I found the smoothness assumption in Theorem 3.7 to be somewhat unnatural. I understand it is a technical necessity at the end, but since the paper aims to introduce an estimator which does not require a-priori smoothness assumptions, I would have expected to see an example which demonstrates this exact point.

---

> ### Author Rebuttal · Authors · 2025-07-31
>
> We sincerely appreciate your detailed and constructive feedback. Below, we address your identified weaknesses (W1–W4) and questions (Q1–Q5).
>
> ## W1
>
> We will revise the discussion following Eq. 7, where the sieved estimator is first introduced, to clarify its main idea as following:
>
> Note that $\varphi^*(\nabla\varphi(x))=\langle x,\nabla\varphi(x)\rangle-\varphi(x)$. By restricting $x$ to $B(0,\tilde{R})$, we equivalently restrict the domain of $\varphi\_{\tilde{R}}^\*(y)$ to the compact set $\nabla\varphi(B(0,\tilde{R}))$. This compactness facilitates us to analyze the estimation error of sieved OT estimator.
>
> Also, we will include more explanations of our crucial ideas (e.g., linking OT estimation error to Brenier potential estimation error) in the revised manuscript.
>
> ## W2 & Q1
>
> We have followed your helpful comments to revise the statements and added more explanation and examples.
>
> Assumption 3.1 depends only on the function class $\mathcal{F}$ and is independent of the true OT map $\nabla \varphi_0$, as well as the measures $\mu$ and $\nu$. It primarily characterizes the regularity of the function class and is essential for controlling the second-order Taylor expansion, which is a key tool in linking the OT estimation error to the Brenier potential estimation error. Additionally, Assumption 3.1 guarantees an envelope function for the candidate function class, which is important for empirical process analysis.
>
> This assumption is not restrictive and is satisfied by many common function classes, including wavelets, RKHS, and neural networks (possibly with engineering modifications such as parameter truncation). For detailed discussion and explicit examples, we refer to Section 4 of [6], whose constructions apply directly to our setting.
>
> Assumption 3.2 quantifies the maximum discrepancy between the estimated OT map $\nabla\varphi$ and the true OT map $\nabla\varphi_0$. This assumption requires that the candidate function class does not include functions that are too distant from $\varphi_0$.
>
> A sufficient condition for Assumption 3.2 is that both Proposition A.2 in Appendix and Assumption 3.1 hold. We clarify that Proposition A.2 allows both $\mu$ and $\nu$ to be heavy-tailed, e.g., Student-t, Pareto distributions. Specifically, Proposition A.2 does not force the density of $\mu$ to be exponentially decay ($V$ can be a logarithmic function), and Definition A.1 accommodates polynomial-tailed $\nu$.
>
> ## W3
>
> Our current analysis, based on empirical process, assumes access to the global minimizer and does not account for optimization error. We will discuss this point more explicitly in the revised manuscript. In addition, we will include explicit bounds on the network complexity, which are originally stated in the Appendix, Theorem E.1.
>
> ## W4
>
> The current state of OT theory relies on the smoothness assumption for approximation error (but not for statistical error and sieve bias). Due to space limitations, we kindly refer you to our response to Q4 from Reviewer mmaX, where we outline the theoretical challenges without such assumptions.
>
> That said, we fully agree it is important to demonstrate how our framework can operate without a-priori smoothness assumptions. As in our response to your Q2, we provide an example of OT rank estimation. Specifically, when the target $\nu$ is the uniform measure on the unit hyper-ball, and the source $\mu$ satisfies some mild conditions, the smoothness of $\varphi_0$ follows automatically from [1], eliminating the need for a-priori assumptions presented in Theorem 3.7. We also refer you to Q4 from Reviewer mmaX on this point.
>
> In conclusion, while smoothness assumptions remain crucial for establishing approximation error, they are unnecessary in some key OT applications. We will revise the manuscript to clarify these distinctions and highlight such examples.
>
> ## Q2
>
> Let us elaborate using the OT-rank problem [2,3] as an example, which provides an important generalization for the notion of ranks to multivariate settings. However, to estimate the OT-rank from $\mu$ to target $\nu$, existing works assume both distributions have compact and convex supports. These assumptions exclude canonical cases, such as unbounded univariate distributions (e.g., Gaussian or Student’s $t$) or the banana-shaped bivariate distributions often used to motivate the OT-rank concept.
>
> When $\nu$ is compactly supported (common for OT-rank to mimic the role of the uniform measure in univariate ranks), the true OT map $\nabla\varphi_0$ must be uniformly bounded regardless of $\mu$. In such cases, any $\mathcal{F}$ with uniformly bounded gradients and Hessians ensures that Assumptions 3.1 and 3.2 hold, with constants $U_1$, $U_2$. Assumption 3.5 is independent of the smoothness. Consequently, Theorem 3.6 holds without a-priori smoothness assumptions.
>
> Thus, with a proper selection of sieve radius $\tilde{R}$ and $\mathcal{F}$, Theorem 3.6 highlights the promise of sieved estimators in improving existing OT-rank estimators.
>
> ## Q3
>
> We first emphasize that Eq. 19, quantifying the sieve bias, has no "pre-factors". It arises solely from the choice of the sieve radius and is independent of $d$.
>
> Regarding Eq. 18, we clarify that $d$ does not appear explicitly, despite implicitly upon $D_\mathcal{F}$ and $M$ (to be explained). It is common for statistical error not to directly depend on $d$ in statistical learning theory. For example, [9] analyzed robust ReLU network estimators using Huber loss and also found that both the statistical error and the bias are dimension-free. The suppressed constant in Eq. 18 depends only on $\gamma$, $\gamma^\prime$, coming from empirical process theory, and is often neglected.
>
> Nevertheless, $D_\mathcal{F}$ and $M$ may depend on $d$. For instance, if $\mathcal{F}$ is a neural network class with Lipschitz activation, $D_\mathcal{F}$ typically scales linearly with the number of neurons. As $d$ increases, achieving a good approximation of $\varphi_0$ may require increasing the network size, causing $D_\mathcal{F}$ to depend on $d$. Similarly, $M$ may scale with $d$ through the term $\sup\\|\nabla\varphi_0(x)\\|\_2$, as shown in Proposition A.2.
>
> In summary, while the statistical error in Eq. 18 is nominally dimension-free, some dimension dependence still arises indirectly via the complexity of $\mathcal{F}$ and regularity properties of $\varphi_0$.
>
> ## Q4
>
> When sieve radii are $\infty$, our sieve estimator reduces to the dual-type estimator studied in prior works [4-7]. In this regime, the sieve bias term $\mathcal{E}_{sieve}$ vanishes, and Proposition 3.4 is the same as in these works.
>
> But our formulation offers a slightly different bound for approximation error, where we express it as $2\inf_{\varphi\in\mathcal{F}}\\|\varphi-\varphi_0\\|\_{L^\infty},$ which connects more directly to classical function approximation theory. In contrast, prior works either represent it with the excess risk $\inf_{\varphi}(\mu\varphi+\nu\varphi^\ast)-(\mu\varphi_0+\nu\varphi_0^\ast);$ or based on the gradient $\inf_{\varphi\in\mathcal{F}}\\|\nabla\varphi-\nabla\varphi_0\\|\_{L^2(\mu)},$ which are harder to analyze.
>
> ## Q5
>
> We will discuss in Section 3.4 to clarify why our framework currently focuses on tanh activation. Indeed, incorporating ReLU networks into the current framework poses two challenges:
> 1. **Gradient-based definition of the OT:** Our estimator is defined via the gradient of a potential function. While PyTorch can handle ReLU networks computationally, the ReLU activation is not globally differentiable in mathematics. So, this definition is ill-posed for ReLU networks.
> 2. **Second-order regularity:** Even if we use a weak derivative to define OT, ReLU networks do not admit second-order weak derivatives. However, existing OT theory fundamentally relies on second-order regularity to perform statistical analysis.
>
> That said, we conducted additional experiments with larger ReLU networks and found that our smaller TNNs with smooth activations still outperformed them (see our response to Reviewer wHrA, comment L1). This aligns with recent findings (e.g., [8] shows that ReQU networks can achieve the same approximation accuracy as ReLU using fewer neurons). These results suggest that smooth activations like tanh may offer advantages in OT estimation, both theoretically and empirically, and may warrant deeper investigation in future work.
>
> ## Typo
>
> Last but not least, thanks to your careful reading, we have also corrected the typo in Proposition A.2.
>
> ## Reference
>
> [1] Del Barrio et al. A note on the regularity of center-outward distribution and quantile functions. JMVA 2020
>
> [2] Chernozhukov et al. Monge–Kantorovich depth, quantiles, ranks and signs. AoS 2017
>
> [3] Ghosal and Sen. Multivariate ranks and quantiles using optimal transport: Consistency, rates and nonparametric testing. AoS 2022
>
> [4] Ding et al. Statistical convergence rates of optimal transport map estimation between general distributions. ArXiv 2412.08064.
>
> [5] Hütter and Rigollet. Minimax estimation of smooth optimal transport maps. AoS 2021
>
> [6] Divol et al. Optimal transport map estimation in general function spaces. AoS 2025
>
> [7] Gunsilius. On the convergence rate of potentials of Brenier maps. ET 2022
>
> [8] Abdeljawad. Uniform approximation with quadratic neural networks. Neural Networks 2025
>
> [9] Fan et al. How do noise tails impact on deep relu networks? AoS 2024

---

> > ### Comment · Reviewer_qHh5 · 2025-08-07
> > **Response**
> >
> > Thank you for the very elaborate response.
> > I maintain my overall positive impression of the paper.

---

### Official Review · Reviewer_wHrA · 2025-06-19

**Clarity:** 3
**Significance:** 3
**Originality:** 3
**Rating:** 4
**Confidence:** 1

**Summary:**

This paper provides a new estimation theory with relaxed assumptions. Specifically, the paper provides a novel map stability inequality and a refined oracle inequality.

**Questions:**

see above

**Ethical Concerns:**

["NO or VERY MINOR ethics concerns only"]

**Final Justification:**

This paper is a technical strong paper. However, its current format may need further polishing. Therefore, I maintain my scores.

**Limitations:**

Results are shown only for tanh neural networks, which are less commonly used in modern deep learning compared to ReLU or more complex architectures.

The theoretical framework is mathematically dense and may be less accessible to non-specialists, which could limit adoption.

**Quality:**

3

**Strengths And Weaknesses:**

Honestly, I am not an expert in optimal transport theory, so I do not feel qualified to make a meaningful judgment on this paper. However, given its technical focus and theoretical depth, I believe it may be better suited for a probability-oriented journal, such as The Annals of Statistics, as the venue of [1].

**Comments and Questions:**

1. The paper should explain what dual-type OT map estimators are at the beginning of the introduction. Additionally, it would help to clarify what is meant by statistical error and approximation error, as these are central to the paper’s contribution.

2. In the Introduction, the authors claim that the assumptions in [27] exclude many practical OT applications. It would be helpful to provide concrete examples of OT applications that are excluded.

3. In Assumption 3.2, what is the envelope function?

4. On line 150, the authors state that Assumptions 3.1 and 3.2 require only that the true OT map $\nabla \varphi_0$ be bounded. This is confusing, since Assumptions 3.1 and 3.2 are stated for all $\varphi \in \mathcal{F}$.

5. In Assumption 3.5, what does the notation $\log_+$ mean? Please define it explicitly.

6. What is the meaning of $\mathcal{O}_{\log}$ notation? It appears nonstandard and would benefit from a formal definition.

7. In the training procedure, how do you enforce that the TNN functional class satisfies equation (23)?


[1] Jan-Christian Hütter and Philippe Rigollet. Minimax estimation of smooth optimal transport
400 maps. The Annals of Statistics, 2021.

---

> ### Author Rebuttal · Authors · 2025-07-31
>
> We sincerely appreciate your thoughtful and encouraging comments on our work.
>
> Our main contributions lie at the intersection of statistics, machine learning, and optimization, making it highly relevant to the NeurIPS community. Our paper is motivated by practical learning problems and designed for algorithmic implementation, which we believe makes NeurIPS the ideal venue for this contribution.
>
> Below, we provide our point-by-point responses to your specific questions (Q1–Q7) and mentioned limitation (L1).
>
> ## Q1
>
> Thank you for this helpful suggestion. We agree that clarifying these foundational concepts to enhance the accessibility and clarity of our manuscript.
>
> To address your first point, we will revise the introduction to more clearly define dual-type OT map estimators. Specifically, we will include the following sentence:
>
> *This paper aims to contribute to this direction by focusing on the dual-type OT map estimators, formally introduced in Section 2, which are based on the characterization of the optimal transport map as the gradient of a convex function (the so-called Brenier potential).*
>
> We will also clarify **statistical error** and **approximation error** explicitly in our updated manuscript:
> 1. **Approximation error** refers to the inherent gap between the true Brenier potential and the best possible approximation within the candidate function class $\mathcal{F}$. This arises because we do not assume that the true Brenier potential $\varphi_0$ lies in $\mathcal{F}$.
> 2. **Statistical error** captures the discrepancy between the estimator and the best possible approximator in $\mathcal{F}$. It reflects the order of magnitude of randomness introduced by the use of finite samples, i.e., the difference between the empirical and population loss functionals.
>
> ## Q2
>
> We appreciate your suggestions. For example, in image generation, generative models often sample from Gaussian white noise as the source distribution $\mu$, which is unbounded and violates compact support assumptions.
>
> As another example, consider $\mu=N(0,1)$ and $\nu=U[0,1]$. The OT map from $\mu$ to $\nu$ is the CDF of $\mu$, which is not $(\alpha,a)$-convex for any $a\geq0$. This is because the corresponding Hessian is not bounded below by 0. Conversely, the OT map from $\nu$ to $\mu$, as the quantile function of $\mu$, is not $(\beta,b)$-smooth for any $b\geq0$, as it diverges at the boundaries of $[0,1]$.
>
> These examples show that existing assumptions on distributions and OT maps, such as compact supports and $(\alpha,a)$-convexity or $(\beta,b)$-smoothness, exclude canonical OT applications in deep generative models and multivariate statistical inference (a.k.a. OT rank and OT quantile). We can incorporate these examples into the revised introduction.
>
> We also refer you to our response to Question 2 from Reviewer qHh5, where we further emphasize how our results provide theoretical backbones for such OT applications, with a focus on OT rank estimation as an example, and theoretical guarantees not covered by existing works.
>
> ## Q3
>
> Thank you for this helpful question.
>
> In Assumption 3.2, the envelope function refers to a real-valued, non-negative function $\Phi \in L^2(\mu)$ that uniformly bounds the absolute values of all functions in the class $\mathcal{F}\cup\{\varphi_0\}$. Specifically, it satisfies $$|\varphi(x)|\leq\Phi(x),\quad\text{for all $x$ and all $\varphi\in\mathcal{F}\cup\\{\varphi_0\\}$}.$$
>
> This function $\Phi$ provides a uniform control over the magnitude of the entire function class $\mathcal{F}\cup\\{\varphi_0\\}$ and plays a key role in applying the empirical process theory. We will clarify this definition explicitly in the revised version.
>
> ## Q4
>
> We appreciate the reviewer’s comment that our original explanation of Assumptions 3.1 and 3.2 lacked clarity. To address this, we have revised the statements of Assumptions 3.1 and 3.2 in the updated manuscript to better reflect their intended meaning.
>
> Assumption 3.1 depends only on $\mathcal{F}$ and is independent of $\varphi_0$, $\mu$ and $\nu$. It is satisfied by many common function classes, including wavelets, RKHS, and neural networks with smooth activation functions and bounded parameters (see Section 4 of [DNP] for a detailed discussion).
>
> Also, we clarify that a **sufficient condition for Assumption 3.2** is that $\\|\nabla\varphi_0(x)\\|_2$ is bounded by some function $L_1(x)$ and Assumption 3.1 holds. Assumption 3.1 provides an envelope for $\\|\nabla\varphi(x)\\|_2$, denoted as $L_2(x)$. Then, Assumption 3.2 follows from the triangle inequality: $\\|\nabla\varphi(x) - \nabla\varphi_0(x)\\|_2 \leq L_1(x) + L_2(x)$. In our appendix, Proposition A.2 shows that a bound for $\\|\nabla\varphi_0(x)\\|_2$ can be obtained under mild distributional assumptions on $\mu$ and $\nu$. We refer to Appendix A for further details and for an explicit example verifying Assumption 3.2.
>
> ## Q5
>
> Thank you for catching this. We define $\log_+(x):=\max\\{\log x,1\\}$, ensuring the quantity remains non-negative and bounded below by 1. We will include this definition in the updated manuscript.
>
> ## Q6
>
> Thank you for raising this point. We use $\mathcal{O}\_{\log}$ as shorthand for “up to logarithmic factors.” For example, $$\mathcal{O}\_{\log}\bigl((n/D\_{\mathcal{F}})^{-1/2}\bigr)$$
> means there exist constants $c_1,c_2>0$ such that the expression is bounded by $$c_1\cdot (n/D\_{\mathcal{F}})^{-1/2}\log(n/D\_{\mathcal{F}})^{c_2}.$$
> We will define this notation explicitly in the revised manuscript to ensure clarity.
>
> ## Q7
>
> Thank you for your interest in the implementation details of our TNN-based estimator. It is important to explain the dependence of the network architecture parameters on the sample size $n$, dimension $d$, and the smoothness $\alpha$ of the target potential $\varphi_0$ in the manuscript.
>
> In all our experimental settings, no explicit truncation of the TNN parameters was required in practice. According to our mathematical analysis (see lines 942–943 and 969 in the supplement), the hyperparameter $\kappa$, which controls the range of network parameters in the TNN class, should be chosen as $$\kappa = O\left(n^{\frac{d(d+(\lfloor\alpha\rfloor+2)^2+4)+2}{2d+4\alpha}}\right).$$
> In our experiments, all underlying potentials $\varphi_0$ are infinitely smooth. Taking the limit $\alpha\to\infty$ yields $\kappa\to\infty$.
>
> To further investigate the effect of truncation, we conducted supplementary experiments in which all network parameters were clamped by $\pm 10$. Interestingly, we observed that truncation has a  negligible impact on TNN performance. Due to space constraints, we report here the L2-UVP and the mean of the maximum parameter magnitude in the $\mathcal{N}(0,1)$ to $t_6$ setting with $d=10$ across 50 repetitions. The results suggest that parameter values remain small and increase slowly with sample size, further supporting that truncation is primarily a theoretical device, and is unnecessary in practice.
>
> ### Normal to t (d=10), No truncation vs Truncation at 10.
> |C|64|128|256|512|1024|2048|
> |-|-|-|-|-|-|-|
> |2 (No)|0.41|0.26|0.14|0.09|0.06|0.06|
> |2 (10)|0.43|0.26|0.14|0.09|0.07|0.06|
> |inf (No)|0.58|0.42|0.23|0.10|0.06|0.06|
> |inf (10)|0.58|0.44|0.25|0.10|0.07|0.06|
>
> ### Normal to t (d=10), Mean of Max Parameter over 50 repetition
> |C|64|128|256|512|1024|2048|
> |-|-|-|-|-|-|-|
> |0|1.19|1.32|1.74|2.39|3.10|4.02|
> |1|1.31|1.55|2.04|2.81|3.62|4.53|
> |2|1.33|1.58|2.06|2.86|3.71|4.55|
> |3|1.34|1.59|2.09|2.86|3.73|4.65|
> |inf|1.357|1.62|2.11|2.85|3.68|4.56|
>
> ## L1
>
> We acknowledge that tanh networks are less commonly used in practice, and bridging theoretical insights with practical architectures remains one of our long-term goals in optimal transport.
>
> However, as discussed in our response to Q5 from Reviewer qHh5, the mathematical formulation of OT maps as gradients of potential functions is incompatible with ReLU networks, which lack second-order (weak) derivatives. Indeed, the second-order regularity of the candidate function class is foundational in current OT theory for conducting statistical analysis.
>
> Beyond such reasons from theoretical analysis, we have also run one setting in our numerical study with larger ReLU networks. Shockingly, they are significantly worse than TNNs:
>
> ### Median L2-UVP, 3-Hidden Layer and 20-neurons ReLU, Uniform to Normal, d = 10.
> |C|64|128|256|512|1024|2048|
> |-|-|-|-|-|-|-|
> |0|0.62|0.43|0.32|0.28|0.23|0.18|
> |1|0.61|0.43|0.34|0.29|0.24|0.19|
> |2|0.59|0.42|0.34|0.29|0.23|0.19|
> |3|0.59|0.41|0.33|0.29|0.24|0.19|
> |inf|0.55|0.41|0.33|0.29|0.24|0.20|
>
> Practically, ReLU is often preferred for its computational efficiency. However, recent findings suggest that smooth activations may improve approximation capabilities. For instance, [Abd] shows that ReQU networks can achieve the same approximation accuracy as ReLU with a square root number of neurons. Combining the theoretical hardness and our shocking empirical evidence, OT estimation may benefit from smooth tanh activation. This observation may motivate further exploration, especially given the more powerful GPUs available today compared to the early DL era (say AlexNet in 2012).
>
> ## L2
>
> Thanks for this comment. We will include more explanations of our crucial ideas (e.g., the pseudo-compact support brought by sieved estimator, linking OT estimation error to Brenier potential estimation error) in the revised manuscript.
>
> ## Reference
>
> [DNP] Vincent Divol, Jonathan Niles-Weed, and Aram-Alexandre Pooladian. "Optimal transport map estimation in general function spaces." The Annals of Statistics 53.3 (2025)
>
> [Abd] Abdeljawad. Uniform approximation with quadratic neural networks. Neural Networks 2025

---

### Official Review · Reviewer_mmaX · 2025-06-26

**Clarity:** 3
**Significance:** 3
**Originality:** 3
**Rating:** 5
**Confidence:** 3

**Summary:**

The paper presents a new error bound on an optimal transport map estimator based on sieved convex conjugate. In particular, they replace the standard conjugate in the semi-dual form of the Kantorovich problem with the sieved counterpart, showing that it helps relax the smoothness/convexity assumption of the true Brenier potential while still achieving stability. The new bound allows the authors to derive the first error bound for the class of tanh-activated deep neural network potentials while only assuming the smoothness of the true potential and the sub-Weibullness of the source distribution (which are mild and can be relaxed). Experiments on synthetic data with non-smooth/non-convex potentials align with the stability bound.

**Questions:**

- What is $t_6$ in the experiments?
- It might be helpful to empirically compare the proposed method with the original dual estimator on cases of smooth/convex potentials, as well as compare their computational speeds since they have two different optimization subproblems.
- The authors should highlight how the new stability bound with relaxed assumptions helps in Section 3.4.
- How does removing the smoothness assumption in Theorem 3.7 affect the error bound? Since the experiments also consider a non-smooth case, a comment on this might be necessary.

**Ethical Concerns:**

["NO or VERY MINOR ethics concerns only"]

**Final Justification:**

Overall, this is a solid paper with both theoretical interest and practical relevance. In the rebuttal, the authors provided additional experiments that address my earlier concerns, which would improve the paper’s quality. I therefore maintain my current score of 5, leaning toward acceptance.

**Quality:**

3

**Strengths And Weaknesses:**

Strengths:
- The paper is well-written, theoretically sound and contains novel results.
- The bound for the class of neural OT estimators brings the proposed method closer to practicality.

Weaknesses
- The tanh activation is known to suffer from vanishing gradient and slow convergence. It might be helpful to investigate this empirically, e.g., increasing the number of layers in the experiments.

---

> ### Author Rebuttal · Authors · 2025-07-31
>
> We sincerely thank you for your thoughtful review, encouraging remarks, and constructive suggestions. We are grateful for your recognition of the theoretical contributions and practical relevance of our work. Below, we provide our point-by-point responses to your specific questions (Q1 to Q4):
>
> ## Weakness
>
> Thank you for raising this insightful point. Our additional experiments show that deeper TNN architectures can learn faster than 2-hidden-layer TNNs, and achieve slightly better ultimate L2-UVP when trained for sufficient epochs. Meanwhile, only minor gradient vanishing issues were observed in our setting.
>
> Due to the time limit, we only consider the setting of normal to uniform, $C=2$, $d = 5$. Five TNNs are compared in the experiment: Model 1 ($f_{2,10}$), Model 2 ($f_{8,10}$), Model 3 ($f_{8,4}$), Model 4 (Model 2 with residual connections), and Model 5 (Model 3 with residual connections), where $f_{D,W}$ denotes $D$ hidden layers and width $W$. The learning rate was set to $0.001$ and each model was trained for 300 epochs with $n = 256$.
>
> We observed that the 8-layer TNN models (Models 2 and 4) demonstrated a faster decrease in loss compared to the 2-layer model. The addition of residual connections offered a slight improvement in learning efficiency, suggesting only a minor vanishing gradient issue in our setting. Throughout training, Models 2 and 4 consistently outperformed Model 1, while Models 3 and 5 performed notably worse.
>
> ### Mean Loss over 50 Repetition
> | epoch | model1 | model2 | model3 | model4 | model5 |
> |-------|--------|--------|--------|--------|--------|
> | 1     | 29.03  | 28.99  | 29.00  | 28.86  | 28.85  |
> | 31    | 17.72  | 11.12  | 19.43  | 6.06   | 19.63  |
> | 61    | 5.80   | 4.86   | 10.33  | 4.09   | 10.17  |
> | 91    | 4.56   | 3.26   | 7.87   | 2.61   | 7.99   |
> | 121   | 3.38   | 2.56   | 6.57   | 2.07   | 6.53   |
> | 151   | 2.48   | 2.11   | 6.76   | 1.81   | 5.42   |
> | 181   | 2.01   | 1.89   | 5.06   | 1.69   | 4.89   |
> | 211   | 1.84   | 1.72   | 4.23   | 1.60   | 4.24   |
> | 241   | 1.75   | 1.63   | 4.15   | 1.53   | 3.92   |
> | 271   | 1.69   | 1.57   | 3.87   | 1.51   | 3.59   |
> | 300   | 1.64   | 1.55   | 3.40   | 1.49   | 3.35   |
>
> ### L2-UVP
> | model1 | model2 | model3 | model4 | model5 |
> |--------|--------|--------|--------|--------|
> | 0.095  | 0.089  | 0.345  | 0.094  | 0.369  |
>
> ## Q1
>
> The notation $t_6$ refers to the Student’s $t$-distribution with 6 degrees of freedom, chosen in line with [DLX]. As a heavy-tailed distribution, it allows us to test the estimator beyond the compact support assumptions dominating the literature.
>
> ## Q2
>
> Thanks for this suggestion.
>
> To compare the sieved estimator with original one on cases of smooth/convex potentials, we conducted additional experiments under settings as in [HR], where the source is a uniform distribution and the true OT map is either the identity or exponential mapping. The following table summarizes the median (standard deviation) of L2-UVP for source = uniform and OT = exp, d=10.
>
> ### Median(std) of L2-UVP, uniform to exp(uniform), d = 10
> | C   | 128           | 256           | 512           | 1024          | 2048          |
> |-----|---------------|---------------|---------------|---------------|---------------|
> | 0   | 0.120 (0.017) | 0.071 (0.009) | 0.047 (0.005) | 0.031 (0.002) | 0.027 (0.004) |
> | 1   | 0.115 (0.020) | 0.065 (0.008) | 0.048 (0.006) | 0.033 (0.004) | 0.028 (0.005) |
> | 2   | 0.119 (0.023) | 0.065 (0.007) | 0.048 (0.006) | 0.033 (0.005) | 0.030 (0.005) |
> | 3   | 0.114 (0.023) | 0.065 (0.008) | 0.048 (0.006) | 0.034 (0.004) | 0.029 (0.005) |
> | inf | 0.122 (0.030) | 0.068 (0.011) | 0.049 (0.005) | 0.034 (0.005) | 0.031 (0.005) |
>
>
> It demonstrates that the sieved estimator consistently outperforms the original one in the setting of compact support and smooth/convex OT maps.
>
> The mean training time between the sieved estimator and the original estimator is summarized below:
>
> ### Mean training time (second), normal to uniform, sieved estimator vs Original Estimator
> | C   | 64      | 128      | 256      | 512      | 1024      | 2048      |
> |-----|---------|----------|----------|----------|-----------|-----------|
> | 2   | 97.8096 | 159.7494 | 293.5467 | 587.9228 | 1107.5363 | 2159.0094 |
> | inf | 85.7766 | 140.1795 | 257.9078 | 513.5465 | 968.8996  | 1890.1076 |
>
> The sieved estimator incurs an additional computational cost of approximately 15\% due to the projection step. However, we believe this increase in computational time is moderate and acceptable, given the practical benefits of the sieved estimator.
>
> ## Q3
>
> We agree that the role of the new stability bound in relaxing assumptions should be emphasized more clearly. We will revise Section 3.4 accordingly.
>
> To highlight the benefits of our new stability bound, our Theorem 3.7 improves the results in [DLX] in several ways:
> 1. **Characterization of Approximation Error.** A crucial conceptual difference lies in how the approximation error is quantified. In [DLX], it is expressed in terms of the excess risk $\inf_{\varphi} (\mu\varphi + \nu\varphi^\ast) - (\mu\varphi_0 + \nu\varphi_0^\ast)$, which does not directly connect with classical approximation theory tools.
> In contrast, we express approximation error in terms of the $L^\infty$ norm $\inf_{\varphi}\\|\varphi - \varphi_0\\|\_{L^\infty(B(0,R\_\varepsilon))}$, which is easier to analyze.
> 2. **Flexibility in Sieve Radius** Our result accommodates a wide range of sieve radii, allowing an adaptive selection based on the data and function class. In contrast, [DLX] fixes the sieve radius to the maximum norm of the source samples, $\max_{1\leq i\leq n}\\|X_i\\|_2$, and their proof relies on this specific choice.
> 3. **Assumptions on Sample Sizes.** While we assume $n = m$ in Theorem 3.7, it is only for theoretical analysis simplicity and can be relaxed (see Thm 3.6 for evidence). In contrast, [DLX] assumes $m \lesssim n$ due to their choice of sieve radius. This assumption appears restrictive and somewhat counterintuitive, as it limits the use of larger sample sizes from $\nu$, which should improve estimation rate. Upon closer inspection, this assumption in [DLX] is tightly coupled with their selection of sieve radius, and is challenging to remove.
>
>
>
> ## Q4
>
> We appreciate the reviewer’s interest in the smoothness assumptions underlying Theorem 3.7.
>
> We impose that $\varphi_0$ satisfies a $(\beta,b)$-smoothness condition to facilitate the approximation error analysis. Removing this $(\beta,b)$-smoothness may degrade the approximation error bound to $O(\varepsilon^a)$ for $a<1$. In principle, one can repeat the proof to derive a convergence rate by balancing this additional exponent $a$ across the statistical error, approximation error, and sieve bias terms. Since the approximation error becomes larger, the overall convergence rates are expected to be slower. We agree with your suggestion and will add a discussion in the revised manuscript to acknowledge this limitation and empirical observation.
>
> Indeed, the $(\beta,b)$-smoothness assumption is widely used in the OT estimation literature (e.g., [Gun]; [HR]; [Man+24]; [DNP]; [DLX]), as it provides a convenient and tractable framework for theoretical analysis. To our knowledge, there is currently no well-developed alternative for systematically characterizing the regularity or approximation properties of OT maps under weaker or more general conditions.
>
> While classical OT theory (e.g., [Caf92]; [Fig17]) offers sufficient conditions for $(\beta,b)$-smoothness based on stringent assumptions on the source and target measures, these conditions are complex and difficult to verify in practice. Incorporating them directly into our framework would substantially complicate both the analysis and exposition, and may ultimately make our work less accessible.
>
> Thus, while the $(\beta,b)$-smoothness assumption introduces theoretical limitations, it remains a standard and necessary compromise in the current state of the literature. Exploring alternative regularity frameworks that can broaden applicability without sacrificing analytical tractability remains an important and challenging direction for future research.
>
> ## Reference
>
> [HR] Jan-Christian Hütter, Philippe Rigollet "Minimax estimation of smooth optimal transport maps," The Annals of Statistics 49.2 (2021): 1166-1194.
>
> [DLX] Ding, Yizhe, Runze Li, and Lingzhou Xue. "Statistical convergence rates of optimal transport map estimation between general distributions." ArXiv 2412.08064.
>
> [DNP] Vincent Divol, Jonathan Niles-Weed, and Aram-Alexandre Pooladian. "Optimal transport map estimation in general function spaces." The Annals of Statistics 53.3 (2025): 963-988.
>
> [Gun] Florian F Gunsilius. "On the convergence rate of potentials of Brenier maps." Econometric Theory 38.2 (2022): 381-417.
>
> [Man+24] Manole, Tudor, et al. "Plugin estimation of smooth optimal transport maps." The Annals of Statistics 52.3 (2024): 966-998.
>
> [Caf92] Luis A Caffarelli. "The regularity of mappings with a convex potential." Journal of the American Mathematical Society 5.1 (1992): 99-104.
>
> [Fig17] Alessio Figalli. The Monge-Ampere equation and its applications. Zürich: European Mathematical Society, 2017.
>
> [Abd] Ahmed Abdeljawad. "Uniform approximation with quadratic neural networks." Neural Networks (2025): 107742.

---

> > ### Comment · Reviewer_mmaX · 2025-08-06
> > **Rebuttal Response**
> >
> > I thank the authors for their responses, which have satisfactorily addressed my concerns. As noted by other reviewers as well, some aspects of the presentation would benefit from further clarification and polishing. The additional experiments provided will also strengthen the paper and enhance its overall contribution. I will maintain my current score.

---

### Official Review · Reviewer_6m4y · 2025-07-03

**Clarity:** 2
**Significance:** 3
**Originality:** 3
**Rating:** 5
**Confidence:** 4

**Summary:**

This paper studies the problem of estimators of optimal transport (OT) maps pushing a  measure $\mu$ to a measure $\nu$. Consider the semi-dual of the Kantorovich OT formulation,  $\min_{\phi \in L^1(\mu)} \int \phi \, d\mu + \int \phi^\ast \, d\nu$, where $\phi^\ast$ is the convex conjugate of $\phi$. The minimizer $\phi_0$ of the above problem corresponds to the OT map via $T = \nabla \phi_0$. The authors propose a _sieved estimator_
\begin{equation*}
    \tilde \phi_{n,m} := \arg \min_{\phi \in \mathcal{F}} \int \phi \, d \mu_n + \int \phi^\ast_R \, d \nu_m
\end{equation*}
where $\mu_n$ and $\nu_m$ are the empirical measures of $\mu$ and $\nu$, with $n$ and $m$ samples, respectively, $\mathcal{F}$ is some class of functions, and the sieved convex conjugate is defined as $\phi^{\ast}_{R}(y) := \inf_{x : \|x\|_2 \leq R} \left \{ < x, y >  - \phi(x) \right \}$.

Under the existence of _envelopes_ for both the function class $\mathcal{F}$ and the target OT map $T = \nabla \phi_0$, the authors prove four main results.

**First**, a stability estimate, i.e., a bound on $\| \nabla \phi - \nabla \phi_0 \|_{L^2}^2$ in terms of the _truncated excess risk_, the classical excess risk associated to the above objective truncated at a radius $R > 0$. The bound also contains an error term, termed _residual risk_, in terms of the tail probability of $\mu$ and the aforementioned envelopes. A trade-off can be seen here as larger $R$ minimize the residual risk but penalize for imperfect minimizers of the truncated excess risk.

**Second**, the authors prove an oracle inequality, i.e., a bound on the truncated excess risk in terms of (a) the statistical error associated with the empirical measures $\mu_n$ and $\nu_m$, (b) the sieve error associated with a finite cutoff radius $0 < R < \infty$, (c) the approximation error $\min_{\phi \in \mathcal{F}} \| (\phi - \phi_0) \, {1}{B_R(0)} \|_{L^\infty}$.

**Third**, non-asymptotic statistical rates are established for the convergence of the statistical error using the oracle inequality above together with an assumption on the $L^\infty$ covering number of the function class $\mathcal{F}$. Typical rates of the order $n^{-1/2} + m^{-1/2}$ are obtained, up to $\log n$ factors, with constants depending on the sieved radius $R$ and the envelope functions. The authors also show how to control the sieve error in terms of the sieve radius and the envelope of $\mathcal{F} \cup \{\phi_0\}$. The derived rates are further used to motivate a choice for $R$ in terms of samples from $\mu$ and $\nu$.

**Fourth**, the authors derive non-asymptotic statistical rates for the convergence of neural network OT map estimators. Specifically, to obtain such an estimator one solves the above sieved semi-dual objective with $\mathcal{F}$ consisting of tanh-activated neural networks (TNNs). Here, the target potential is in $C^\alpha(\mathbb{R}^d)$ and the derived rate is $n^{-\frac{\alpha}{2\alpha + d}}$.

The authors conduct experiments using the above framework to learn OT maps that are either non-smooth or non-convex. The TNN estimator disused above is used. An "$L^2$ unexplained variance" objective is used to evaluate the performance of the estimator; $L^2$-UVP is plotted against the number of samples $n$.

It is notable that the the first three results above, i.e. the stability estimate, the oracle inequality and the convergence of the statistical error, only depend on the existence of envelope functions for $\mathcal{F}$ and $T$. In contrast to previous work, no explicit assumptions on the tail behavior or the smoothness or convexity of the OT map are made. However, the convergence results on the TNN estimator require a $(\beta, b)$-smooth target potential and a sub-Weibull source measure $\mu$.

**Questions:**

Could the authors please:

1. Address a potential typo in Assumption 3.1: the function $U_2$ is not assumed to belong to any function class making the assumption, as stated, vacuous. Is the assumption that $U_2$ is point-wise finite?

2. Give examples of function classes $\mathcal{F}$ and measures $\mu, \nu$ such that Assumptions 3.1 and 3.2 are satisfied. It is not necessary, but recommended, to also give examples where the $\mathcal{F}$ contains the target potential $\phi_0$.

3. Further discuss how to lift the smoothness assumption on the target potential in the TNN estimator results. Also, discuss significance of the sub-Weibull assumption on the source measure and why that is not limiting.

4. Discuss why the experiments do not demonstrate the claimed rates.

5. Address these questions on the proofs:
- In the proof of Proposition 3.3 why is it that $D_x \geq 1$? How do you avoid degenerate cases where $U_2 \equiv 0$, for example.
- In the conclusion of line 872 how is equation (42) used? It seems that this follows directly from (43) and the fact that $\mu$ is a probability measure.
- In line 930 you claim to be using Chebyshev but the bound below below line 929 is on the mean not the variance. Please clarify.
- What is capital $N$ appearing in the statement of Theorem 3.7 in the appendix? It does not appear on the original formulation of the paper.
- To obtain the relations below line 947 you are using the face that $\phi_0$ is $(\beta, b)$ not that it is in $C^\alpha$.
- The term $\langle R_\epsilon \rangle$ appears quite a lot but it is an abuse of notation: you defined $\langle x \rangle = \|x\| + 1$ but $R_\epsilon$ is a scalar, not a vector. Mention this.
- What is the role of the expression $R_\epsilon = K (\log 2/\epsilon)^{1/\lambda}$ in the proof of Theorem 3.7? I am assuming this relationship is important because once $\epsilon$ is chosen to be polynomial in $n$ then $R_\epsilon$ is logarithmic in $n$ and so it can be suppressed. If so, then say this. This is important as it is the manifestation of the sub-Weibull assumption.
- What is $\gamma'$ in the display below line 968? I thought that by combining the display below line 966 with assumption 3.5 we have $\gamma = 0, \gamma' = 1$.
- In the statement of Lemma F.2 the right hand side should be $\mathcal{F}$ not $\bar{\mathcal{F}}$. The same mistake appears in the proof. Also in line 960 it should be $h/6F$ not $h/6B$.

6. Other typos, just FYI:
-  Display on line 855 two commas in $B(0, R_\epsilon)$.
- Line 863 Brenier should be capitalized.
-  Line 843 ``the'' is missing.
-  In equations (26-28) I am assuming the $\nu \, f$ is meant to mean $\int f \, d\nu$? Look below as to why this is bad notation.
- Line 881 instead of "the the" it should be "the same."
- Your final result under line 882 involves $\sup_{g \in \bar{\mathcal{G}}} \int_{\nabla \phi_0 (B(0, R_\epsilon))} \ldots$ where the integral is over the image of the ball $B(0, R_\epsilon)$ under the OT map $\nabla \phi_0$. However, the stated result in the main text has the same integral but over the ball $B(0, R_\epsilon)$. Please reconcile this.
- If $\log_+ = \max\{0, \log\}$ then equation (52) has a mistake: the upper bound in the integral should be $1$ not $e$.


7. Other suggestions:
- For a measure $\mu$ avoid the shorthand $\mu \, f$ for the integral $\int f \, d\mu$. This looks like a scalar multiplying a function. Instead, opt for $\langle \mu, f \rangle $  which is way more standard in analysis, specifically distribution theory.
- Be more explicit when using non-trivial results. For example, in equation (31) say you are using the Fenchel-Young inequality.
- Line 926: ``since $\gamma' \geq 1$ _and_ $D_{\mathcal{F}} > 1$, re-organizing \dots''


I am ready to raise my overall rating if Questions 2--4 above are addressed, especially Question 2.

**Ethical Concerns:**

["NO or VERY MINOR ethics concerns only"]

**Final Justification:**

I am quite satisfied with the authors' response to my earlier concerns, and I am now convinced that this paper provides useful contributions to the broader statistical theory of optimal transport map estimation. I am therefore raising my score.

**Limitations:**

Yes.

**Paper Formatting Concerns:**

None.

**Quality:**

3

**Strengths And Weaknesses:**

**Strengths**
- This is an interesting paper with mostly well-structured mathematical arguments.
- The goal is clear and valuable to the statistical OT community.
- The tools used are fitting and the results are interesting and intuitive---for example, the trade-offs between the truncated excess risk and the residual risk or the scaling of the sieve radius with the sub-Weibull parameter $\lambda$ of the source measure, in Theorem $3.6$.
- Moreover, the paper gives what might be the (?) first non-asymptotic statistical rates for a neural OT map estimator, a significant contribution to the literature.

**Weaknesses**

The paper has two major and one minor weaknesses.
- First, Assumptions $3.1$ and $3.2$, i.e. the existence of the envelope functions, are quite opaque. It is a priori not clear what function classes admit envelope functions or for what measures $\mu, \nu$ the OT map $T$ admits an envelope. Now if the assumptions are satisfied, the transport map $|\nabla \phi_0$ is bounded by the envelope functions.  In Appendix A, the authors demonstrate that such a bound holds for a large class of OT maps, namely maps between $\mu \propto e^{-\frac{\|x\|^{2p}}{2}}$ and $\nu$ with polynomial tails.
However, this reviewer maintains that this is a _necessary_ and not a _sufficient_ condition for the assumptions to hold. Here, we would like to see examples of $\mu$, $\nu$, _and_ $\mathcal{F}$ such that the assumptions hold.

- Second, the TNN estimator result does, in fact, assume a smooth target potential and a sub-Weibull source measure. The authors argue that the smoothness assumption is for "convenience" since it is used to "only bound the approximation error and not the statistical rate."
We would like this to be discussed further.

- Third, the experiments merely show that the sieved TNN estimator outperforms the un-sieved TNN estimator, but do not demonstrate the claimed rates.

---

> ### Author Rebuttal · Authors · 2025-07-31
>
> We sincerely thank you for your thoughtful review and constructive comments. Below, we respond to your specific questions (Q1 to Q7):
>
> ## Q1
> Thanks for your comment. We will revise Assumption 3.1 to improve its clarity as follows:
>
> **Assumption 3.1 (Revised)** For function class $\mathcal{F}\subset C^2(\mathbb{R}^d)$, suppose there exists a non-decreasing and pointwise finite function $U_2:[0,\infty)\to[1,\infty)$ such that: $$\sup_{\varphi\in\mathcal{F}}\sup_{\\|x\\|\leq R}\\|\nabla^2 \varphi(x)\\|\_{op}\leq U_2(R),\quad\sup_{\varphi\in\mathcal{F}}\\|\nabla\varphi(0)\\|\_2\leq U_2(R).$$
>
> We explain the roles of the lower and upper bounds of $U_2$:
>
> The upper bound of $U_2$ is critical for the stability inequality in Eq. 14. If $U_2(R)=\infty$ for some $R>R_0$, the RHS of Eq. 14 is $\infty$. So we require $U_2$ to be pointwise finite.
>
> The lower bound $U_2\geq 1$ is a technical assumption. To prove Proposition 3.3, we require a quadratic upper bound on $\varphi(z)$ on L846. In degenerate cases where $\sup_{\varphi\in\mathcal{F}}\sup_{\\|x\\|\leq R}\\|\nabla^2\varphi(x)\\|\_{op} =0$ (say $\mathcal{F}$ consists of linear functions), any positive constant for $U_2$ suffices to ensure that the argument remains valid. Since such constants do not impact convergence rates, we impose $U_2(\cdot)\geq1$ to rule out vacuous bounds.
>
> ## Q2 &  W1
> Thanks for the constructive feedback. We now clarify how Assumptions 3.1 and 3.2 can be satisfied and provide concrete examples where they hold. Notably, they play a role similar to Assumption A.1 in [1], and are strictly weaker. As a result, the function classes discussed in Section 4 of [1] apply directly to our setting.
>
> Assumption 3.1 depends only on $\mathcal{F}$ and is independent of $\varphi_0$, $\mu$ and $\nu$. It is satisfied by many common function classes, including wavelets, RKHS, and neural networks with smooth activation functions and bounded parameters (see Section 4 of [1] for a detailed discussion).
>
> Also, we clarify that a **sufficient condition for Assumption 3.2** is that both Proposition A.2 and Assumption 3.1 hold. Specifically, Proposition A.2 ensures $\\|\nabla\varphi_0(x) \\|\_2$ to be bounded by some function $L_1(x)$. Meanwhile, Assumption 3.1 provides an envelope for $\\|\nabla\varphi(x)\\|\_2$, denoted as $L_2(x)$. Then Assumption 3.2 follows via the triangle inequality $\\|\nabla\varphi(x)-\nabla\varphi_0(x)\\|\_2\leq L_1(x)+L_2(x)$.
>
> Typical cases where Proposition A.2 holds include: $\mu$ can be normal or Student-t, and $\nu$ can be any measure with certain tail decay. We refer to Appendix A for further details and an explicit example verifying Assumption 3.2.
>
> Moreover, there are several settings where $\varphi_0\in\mathcal{F}$ and Assumption 3.2 holds automatically.
>
> A canonical example is the class of **quadratic functions**: $$\mathcal{F}\_{quad}=\\{x\mapsto x^{\top}Bx+\langle b,x\rangle:B\in\mathbb{S}_{+}^{d},\\|B\\|\_{op}\leq r_1, \\|b\\|\_2\leq r_2\\}.$$
> When $\mu$ and $\nu$ are Gaussian (or from the same elliptical family), the Brenier potential is quadratic. This case arises in applications such as distributionally robust estimation of covariance and precision matrices.
>
> Another example $\mathcal{F}$ is the class of **convex functions with uniformly bounded Lipschitz constants**, studied in [2]. For instance, if $\nu$ is the uniform distribution on the unit ball, and the support of $\mu$ is open and convex [3], then $\varphi_0\in\mathcal{F}$ regardless of the tail behavior of $\mu$. This setting arises in OT-rank estimation [4, 5].
>
> We will incorporate these examples and clarifications.
>
> ## Q3 & W2
> Thanks for highlighting these important points. Notably, the $(\beta,b)$-smoothness is widely used in the OT literature [1, 8-11], and the sub-Weibull assumption is also standard [1，7，8]. Both assumptions are used in our work primarily for analytical convenience, not because they are fundamentally required for the methodology.
>
> Our refined approximation result (Lemma F.4) relies on bounding the Sobolev norm $\\|\varphi_0\\|\_{W^{2,\infty}([-\tilde{R}\_\varepsilon,\tilde{R}\_\varepsilon]^d)}$, and the $(\beta,b)$\-smoothness assumption ensures that this norm grows polynomially in $\tilde{R}\_\varepsilon$. Meanwhile, the sub-Weibull $\mu$ guarantees that $R\_\varepsilon,\tilde{R}\_\varepsilon$ scale polynomially in $\log(1/\varepsilon)$. Together, these two results ensure that the relevant Sobolev norm is bounded by a polynomial in $\log(1/\varepsilon)$, and the consequent approximation error has the order of $\varepsilon$.
>
> If $(\beta,b)$-smoothness were lifted, one could still obtain approximation bounds, but they may degrade to $O(\varepsilon^a)$ for some $a<1$. This, in turn, complicates the bias-variance trade-off in the statistical analysis by introducing additional terms in the statistical error, approximation error, and sieved estimation bias trade-off. Thus, while not strictly necessary, the smoothness assumption simplifies the proof and keeps the convergence rate clean and interpretable.
>
> The sub-Weibull assumption on source distribution relates to statistical error, approximation error, and sieved estimation error. Generally, a heavier tail in $\mu$ makes these errors larger. The sub-Weibull tail ensures that any potential enlarging factors remain logarithmic in $n$. Importantly, this assumption is not limiting. Specifically, with polynomially tails, $R_\varepsilon$ scales polynomially in $1/\varepsilon$. While the proofs follow similar steps, the resulting convergence rate becomes much more complex. See Equation 3.10 in [8] for an illustration of the added technical complications.
>
> We will incorporate this discussion into the revised manuscript to clarify the roles and implications of both assumptions.
>
> ## Q4 & W3
> Thanks for bringing up this important question.
>
> To clarify, we present our observed convergence rates in experiments for $d=10$, $n\geq 256$, by computing the slopes of logarithm of $L^2$-$UVP$ w.r.t to $\log n$:
>
> |Source/Target|0|1|2|3|inf|
> |-|-|-|-|-|-|
> |normal/t|-0.38|-0.31|-0.31|-0.32|-0.38|
> |uniform/normal|-0.26|-0.22|-0.21|-0.20|-0.19|
> |t/normal|-0.51|-0.53|-0.52|-0.52|-0.52|
> |normal/uniform|-0.70|-0.69|-0.72|-0.71|-0.70|
>
> Two conclusions can be made on this table:
>
> 1. Across all settings, the observed rates exceed the rate $n^{-2/d} = n^{-0.2}$ for the plug-in estimator in [7], suggesting that our OT estimator effectively benefits from the smoothness of the TNN architecture.
>
> 2. In the normal/uniform setting, the rates are much faster than $n^{-1/2}$, suggesting that optimization error dominates the overall convergence. As further evidence, when repeating experiments for $C=2$ and $C=\infty$ with larger sample sizes ($n=1024, 2048, 3064, 4096, 5120$), the exponents become $-0.426$ and $-0.436$, respectively.
>
> Similar phenomena have also been reported in Section 6.2 of [9].
>
> While our experiments do not fully demonstrate the claimed theoretical rates, providing a complete explanation of OT estimation rates in practice is challenging and remains an open question. The numerical results are influenced by many factors, including neural network architectures, optimization methods, and other implementation choices.
>
> ## Q5-Q7
> We appreciate your attention to detail and will ensure all these corrections and clarifications are made in the revised manuscript.
>
> 5.1 Please refer to our response to Q1 & W1.
>
> 5.2 Eq. 42 provides the upper bound $\nu\_\varepsilon(\bar\varphi\_{\tilde{R}\_\varepsilon}^*-\varphi\_0^\*)\leq\\|\bar\varphi-\varphi\_0 \\|\_{L^\infty(B(0,\tilde{R}\_\varepsilon))}$
>
> Eq. 43 gives the lower bound $\nu\_\varepsilon(\bar\varphi\_{\tilde{R}\_\varepsilon}^*-\varphi\_0^\*)\geq-\\|\bar\varphi-\varphi\_0\\|\_{L^\infty(B(0,\tilde{R}\_\varepsilon))}$. L872 follows by taking absolute values.
>
> 5.3 The RHS in L930 is based on Chebyshev inequality and $Var(2\Phi(X) \mathbb{I}(\\|X\\|\_2>R_\varepsilon))\leq E(4\Phi^2(X)).$
>
> 5.4 Introduced in [6], $N$ is the number of subdivisions per coordinate to localize the neural network approximation. It is fundamental to characterize the size of the TNN class.
>
> 5.6 We define $\langle R\rangle=|R|+1$ for $R \in \mathbb{R}$.
>
> 5.7 The sub-Weibull $\mu$ ensures $R_\varepsilon,\tilde{R}\_\varepsilon$ are $O(\log(1/\varepsilon)^{1/\lambda})$. By taking $\varepsilon=O(n^{-\frac{2\alpha}{d+2\alpha}})$, $R_\varepsilon$ is logarithmic in $n$, so dominated by the polynomial terms in $n^{-1}$, and is suppressed. Please refer to our response to Q3 for a detailed discussion.
>
> 5.8 L968 directly follows by Eq. 47. By plug-in $\gamma=0$ and $\gamma'=1$ as in L966, and combining Eq. 66, we then get Eq. 69.
>
> 6.7 We define $\log_+(x)=\max\\{\log x,1\\}$.
>
> All the other typos and suggestions are addressed.
>
> ## Reference
> [1] Divol et al. Optimal transport map estimation in general function spaces. Annals of Statistics (AoS) 2025
>
> [2] Kur et al. Convex regression in multidimensions: Suboptimality of least squares estimators. AoS 2024
>
> [3] Barrio et al. A note on the regularity of center-outward distribution and quantile functions. JMVA 2020
>
> [4] Chernozhukov et al. Monge–Kantorovich depth, quantiles, ranks and signs. AoS 2017
>
> [5] Ghosal and Sen. Multivariate ranks and quantiles using optimal transport: Consistency, rates and nonparametric testing. AoS 2022
>
> [6] Ryck et al. On the approximation of functions by tanh neural networks. Neural Networks 2021
>
> [7] Deb et al. Rates of estimation of optimal transport maps using plug-in estimators via barycentric projections. NeurIPS 2021
>
> [8] Ding et al. Statistical convergence rates of optimal transport map estimation between general distributions. ArXiv 2412.03722
>
> [9] Hutter and Rigollet. Minimax estimation of smooth optimal transport maps. AoS 2021
>
> [10] Gunsilius. On the convergence rate of potentials of Brenier maps. Econometric Theory 2022
>
> [11] Manole et al. Plugin estimation of smooth optimal transport maps. AoS 2024

---

> ### Author Response · Authors · 2025-08-05
>
> Dear Reviewer,
>
> We hope this message finds you well. We wanted to kindly follow up to see if you might have a chance to review our earlier response to your comments. Your feedback is very valuable for us in further improving the manuscript, and we would greatly appreciate any additional thoughts you could share.
>
> Thank you for your time and consideration.

---

> ### Comment · Reviewer_6m4y · 2025-08-06
>
> I thank the authors for their substantial and thorough efforts responding to my earlier comments, both on the larger ideas in the paper and on various technical details. I am quite satisfied with their responses and I think this paper will be of value to the community. I will improve my rating accordingly. My main closing remark is that providing concrete examples of function classes and measures that instantiate Assumptions 3.1 and 3.2—as the authors promise to do—will be very helpful, even if these are essentially drawn from Divol et al.

---

> > ### Author Response · Authors · 2025-08-07
> >
> > We sincerely thank the reviewer for their thoughtful remarks, appreciation of our revisions, and for improving their rating. We also appreciate the suggestion to include concrete examples that instantiate Assumptions 3.1 and 3.2, which we agree will enhance clarity and accessibility.
> >
> > In our revised manuscript, we will add the following illustrative examples and discussions:
> >
> >
> > Assumption 3.1 depends only on $\mathcal{F}$ and is independent of $\varphi_0$, $\mu$ and $\nu$. It is satisfied by many common function classes, for example,
> >
> >
> > 1. Quadratic function class: When $\mu$ and $\nu$ are both Gaussian (or from the same elliptical family), the Brenier potential is quadratic and lies in the following quadratic family:
> >     $$\mathcal{F}\_{quad}=\\{x\mapsto x^{\top}Bx+\langle b,x\rangle:B\in\mathbb{S}_{+}^{d},\\|B\\|\_{op}\leq r_1, \\|b\\|\_2\leq r_2\\}.$$
> >
> > 2. Smooth Neural Networks:
> >     Neural network function classes with all parameters bounded and activation functions that are $C^2$-smooth, such as sigmoid, tanh, or softmax.
> >
> > 3. Reproducing Kernel Hilbert Spaces (RKHS):
> >    Let $K:\mathcal{X} \times \mathcal{X} \to \mathbb{R}$ be a positive-definite kernel on its domain $\mathcal{X} \times \mathcal{X}$, and let $\mathcal{H}\_K$ be the corresponding RKHS with norm $\\| \cdot \\|\_{\mathcal{H}_K}$. We assume $\mathcal{F}$ is $C^4$ and take $\mathcal{F}$ to be the unit ball of $\mathcal{H}_K$, i.e.,
> > $\mathcal{F} = \\{ \varphi \in \mathcal{H}\_K : \\| \varphi \\|\_{ \mathcal{H}\_K} \leq 1 \\}. $
> >
> > Assumption 3.1 and 3.2 together play a role similar to Assumption A.1 in [1], but are strictly weaker. As a result, the function classes discussed in Section 4 of [1], including parametric family, wavelet expansions, RKHS, and Barron Spaces, apply directly to our setting. We refer readers to Section 4 of [1] for a detailed discussion.
> >
> > A sufficient condition for Assumption 3.2 is that both Proposition A.2 (in Appendix A) and Assumption 3.1 hold. Specifically, Proposition A.2 ensures $\\|\nabla\varphi_0(x) \\|\_2$ to be bounded by some function $L_1(x)$ under mild distributional assumptions on $\mu$ and $\nu$. Meanwhile, Assumption 3.1 provides an envelope for $\\|\nabla\varphi(x)\\|\_2$, denoted as $L_2(x)$. Then Assumption 3.2 follows via the triangle inequality $\\|\nabla\varphi(x)-\nabla\varphi_0(x)\\|\_2\leq L_1(x)+L_2(x)$.
> >
> > Proposition A.2 holds in typical cases, such as when $\mu$ is normal or Student-t, and $\nu$ can be any measure with certain tail decay. We refer to Appendix A for further details and an explicit example verifying Assumption 3.2.
> >
> >
> >
> > ## Reference
> > [1] Divol et al. Optimal transport map estimation in general function spaces. Annals of Statistics (AoS) 2025

---

### Note · Authors · 2025-08-11

We thank the Reviewers and Area Chairs for their time and effort. We first outline our key contributions, then summarize our responses.

# Key Contributions

**Stability inequality for sieved estimator.** We introduce a new map stability inequality that controls estimation error of sieved OT map estimators via a truncated excess risk, without assuming second-order regularity of the Brenier potential. It also relaxes compact/convex support assumptions on the distributions.

**Refined oracle inequality.** We derive an oracle inequality that cleanly decomposes estimation error into statistical error, sieve bias, and approximation error.

**Neural OT rates under general distributions.** Leveraging the above tools, we obtain the first non-asymptotic rates for neural OT map estimators between general (including heavy-tailed) distributions.

**Empirical validation.** Numerical experiments confirm the practical benefits of neural sieved estimators.

# Reviewer-Specific Responses

**Reviewer 6m4y.** We clarified Assumptions 3.1–3.2 with concrete examples and detailed the role of sub-Weibull and $(\beta,b)$-smoothness in analyzing neural OT estimators. We also reported empirical slopes for convergence rates.

**Reviewer mmaX.** We added experiments that vanishing gradient issues are minor in our setting and that the sieved estimator consistently outperforms the original estimator in smooth/convex cases. We also revised Section 3.4 to highlight the advantages of our stability bound and discussed the implications of the smoothness assumptions.

**Reviewer wHrA.** We refined notation and terminology, revised Assumptions 3.1–3.2, and provided concrete examples of OT applications excluded in prior literature to emphasize the wider applicability of our results. We also clarified that the assumptions on TNNs are satisfied in our experiments and supported with numerical results.

**Reviewer qHh5.** We clarified the motivation for the sieved estimator, explained Assumptions 3.1–3.2 with examples, and discussed the regularity assumptions and implications of our theoretical results. Also, we explained why existing OT theory cannot directly accommodate ReLU networks.

**Reviewer oQWg.** We presented the practical implementation of the sieved estimator, highlighted how our convergence guarantees extend beyond prior work, and discussed extensions of our framework to other function classes.

We believe our responses address all concerns and clearly highlight our contributions.

---

### Decision · Program_Chairs · 2025-09-17

**Decision:**

Accept (poster)

**Comment:**

This paper develops new theoretical guarantees for optimal transport map estimation under minimal regularity assumptions. All reviewers agree that it is a solid and valuable contribution, and that the paper should be accepted.

The work has many merits. It addresses a timely and important problem, and its theoretical contributions are both strong and original. In particular, it establishes novel stability results and a clean oracle inequality that aligns well with classical approximation theory. The mathematical framework is well-structured, the proofs are both fitting and intuitive, and the results are likely to be useful not only to the statistical OT community but also to applied work where neural estimators are employed. The rebuttal was detailed and constructive, addressing earlier concerns and including additional experiments that further reinforce the contribution.

Some weaknesses were noted. The writing is highly technical, which may make the paper less accessible to non-specialists, and certain assumptions could be presented more transparently. The empirical validation, while helpful, remains limited, focusing mainly on improvements of the sieved TNN estimator over the unsieved version without broader benchmarking. In addition, questions remain about the practical suitability of the sieved estimator, and further discussion would strengthen the paper. Most of these concerns were clarified during the rebuttal, and the authors have committed to improving the exposition and clarifying assumptions in the final version, which should broaden the paper’s impact.

In sum, this is a theoretically strong and timely contribution that advances the understanding of optimal transport map estimation. I recommend acceptance.